# Induced p53 loss in mouse luminal cells causes clonal expansion and development of mammary tumours

Luwei Tao[1,2,*], Dongxi Xiang[1,2,*], Ying Xie[1,2], Roderick T. Bronson[3] & Zhe Li[1,2]

Most breast cancers may have a luminal origin. *TP53* is one of the most frequently mutated genes in breast cancers. However, how p53 deficiency contributes to breast tumorigenesis from luminal cells remains elusive. Here we report that induced p53 loss in *Krt8*[+] mammary luminal cells leads to their clonal expansion without directly affecting their luminal identity. All induced mice develop mammary tumours with 9qA1 (*Yap1*) and/or 6qA2 (*Met*) amplification(s). These tumours exhibit a mammary stem cell (MaSC)-like expression signature and most closely resemble claudin-low breast cancer. Thus, although p53 does not directly control the luminal fate, its loss facilitates acquisition of MaSC-like properties by luminal cells and predisposes them to development of mammary tumours with loss of luminal identity. Our data also suggest that claudin-low breast cancer can develop from luminal cells, possibly via a basal-like intermediate state, although further study using a different luminal promoter is needed to fully support this conclusion.

[1] Division of Genetics, Department of Medicine, Brigham and Women's Hospital, Boston, Massachusetts 02115, USA. [2] Department of Medicine, Harvard Medical School, Boston, Massachusetts 02115, USA. [3] Rodent Histopathology, Harvard Medical School, Boston, Massachusetts 02115, USA. * These authors contributed equally to this work. Correspondence and requests for materials should be addressed to Z.L. (email: zli4@rics.bwh.harvard.edu).

Breast cancer is genetically and clinically heterogeneous[1]. Based on microarray expression profiling, breast cancer can be categorized into several intrinsic subtypes, including basal-like, HER2-enriched, normal-like, luminal A and B, and claudin-low subtypes[2–7]. Among them, the basal-like and claudin-low breast cancers largely overlap with triple-negative breast cancers (TNBCs) defined based on histology. As different breast cancer subtypes have distinct mutation patterns and clinical outcomes, a better understanding of how recurrent mutations contribute to each subtype is essential for unveiling novel disease mechanisms and therapeutic targets. Although mutations in the p53 gene (TP53 in human, Trp53 in mouse) have been found in all breast cancer subtypes, they are particularly prevalent in TNBCs: for instance, TP53 mutations (largely inactivation mutations) have been found in at least 80% of basal-like breast cancer (BLBC) cases[2]; in claudin-low breast cancer, the 17p13.1-p12 genomic region, which harbours TP53, is also frequently deleted[8].

In murine models, constitutive loss of p53 in mammary epithelial cells (MECs) leads to the development of heterogeneous mammary tumours[9]. In preneoplastic murine mammary lesions, mutations of Trp53 can also be frequently detected[10]. In human breast cancer, statistical reconstitution of its natural history based on next-generation sequencing data suggested that TP53 mutation is among the earliest driver mutations that lead to transformation of breast cells[11]. Importantly, a recent genomic study of human High-Grade Ductal Carcinoma In Situ (DCIS), which is believed to be a precursor lesion of invasive ductal carcinoma, revealed that the p53 pathway was deactivated in all DCIS samples analysed, regardless of TP53 mutation status or intrinsic subtypes[12]. A respective analysis of the TCGA invasive breast cancer data set further revealed that inactivation of the p53 pathway is extremely common, affecting >85% of all breast cancer cases regardless of intrinsic subtypes (and TP53 mutation status)[12]. Together, these data suggest that p53 deficiency likely plays a key role in the early stage of breast tumorigenesis.

MECs are organized in a hierarchy. Studies based on cleared fat pad transplantation and lineage-tracing approaches demonstrated that multipotent mammary stem cells (MaSCs), which are basal MECs, could produce both luminal and basal MECs upon transplantation or during development[13–15]. Although basal MaSCs were initially proposed as cells of origin of breast cancer, recent studies demonstrated that luminal MECs might be cells of origin of most breast cancers (including BLBC)[16–20]. In this study, we asked how induced loss of p53 in the luminal lineage affects luminal MECs and whether this leads to development of heterogeneous mammary tumours with a luminal origin. We show that induced loss of p53 in luminal MECs leads to their clonal expansion without directly affecting their luminal fate, but predisposes luminal cells to the development of mammary tumours with loss of luminal identity and acquisition of MaSC-like properties. Thus, our data suggest that although p53 does not dictate the identity of luminal MECs directly, it safeguards them from aberrant proliferation, cell fate alteration, and development of mammary tumours with loss of their original luminal MEC properties.

## Results

### Induced p53 loss in luminal cells leads to clonal expansion.
To study how induced loss of p53 affects luminal MECs, we utilized a conditional approach we developed recently, by intraductal injection of Cre-expressing adenovirus under the control of the pan-luminal Keratin 8 (that is, Krt8 or K8) promoter (Ad-K8-Cre) to nulliparous female mice carrying Trp53 conditional knockout alleles (Trp53$^{L/L}$) and a conditional Cre-reporter,

Rosa26-LSL-YFP (R26Y) (that is, Trp53$^{L/L}$;R26Y females)[21]. Simultaneous disruption of Trp53$^{L}$ and activation of R26Y by Cre leads to genetic marking of Trp53-null luminal MECs by YFP (Fig. 1a). We showed previously that intraductal injection of Ad-K8-Cre to mammary glands (MGs) of R26Y females led to genetic marking of luminal MECs[21]. To further determine the types of luminal MECs targeted by Ad-K8-Cre, we analysed MECs 3 days after injection. By co-immunofluorescence (co-IF) staining, we found that Cre-expressing (from the internalized Ad-K8-Cre viruses) cells were localized in mammary ducts and largely overlapped with YFP-marked MECs (Fig. 1b). It was shown recently that the mammary luminal lineage could be separated as the Sca1$^{-}$CD49b$^{+}$ oestrogen receptor (ER)$^{-}$ luminal progenitor (LP), Sca1$^{+}$CD49b$^{+}$ ER$^{+}$ LP and Sca1$^{+}$CD49b$^{-}$ non-clonogenic luminal (NCL, which is enriched with ER$^{+}$ mature luminal (ML) cells) subpopulations, based on fluorescence-activated cell sorting (FACS) analysis[22]. By FACS using these markers, we found that all three luminal subpopulations could be targeted by Ad-K8-Cre and among them, approximately half of YFP-marked luminal MECs were ER$^{+}$ cells (Fig. 1c,d). To complement the Ad-K8-Cre-based approach, we also generated K8-CreER;Trp53$^{L/L}$ female mice in which injection of tamoxifen activates the Cre-oestrogen receptor (CreER) fusion[15], leading to induced disruption of Trp53 in luminal MECs (Fig. 1a). These two conditional approaches produced similar results.

Based on lineage tracing, multiple studies, including ours, demonstrated that the mammary luminal lineage in adult mice is largely sustained by lineage-restricted luminal stem cells or long-lived progenitors[15,21,23,24]. Interestingly, certain oncogenic events, such as ectopic expression of the PIK3CA$^{H1047R}$ mutant or ETV6–NTRK3 fusion, could induce multipotency in luminal MECs, leading to emergence of basal-like MECs from luminal cells[24–26]. Since TP53 mutations are most frequently associated with BLBC and p53 has been shown to restrict mammary stem/progenitor cells[27–29], we first asked whether induced loss of p53 in the luminal lineage affects self-renewal of luminal MECs and whether this also leads to a similar luminal-to-basal transition phenotype.

On Cre-mediated inactivation of Trp53 and activation of YFP-marking (Pulse), we analysed MGs from the injected females after different chasing time periods. Three days after injection, by co-IF staining, we found that the majority of YFP-marked cells from both R26Y-only control females and Trp53$^{L/L}$;R26Y experimental females were individual luminal cells in mammary ducts (that is, single-cell clones, Fig. 1e, left). Interestingly, 3–4 weeks after injection (that is, short-term chase), we found that in mammary ducts, while the majority of YFP-marked wild-type (WT) MEC clones remained as single-cell clones, many YFP-marked Trp53-null clones contained two or more YFP$^{+}$ cells (Fig. 1e, middle). We quantified percentages of YFP$^{+}$ single-cell clones and multi-cell clones and found that 3 days post injection, there was no significant difference in percentages of these two types of clones when comparing experimental mice with control mice; however, on the 3–4-week short-term chase, we found that significantly more YFP$^{+}$ clones in the injected Trp53$^{L/L}$;R26Y females were multi-cell clones (Fig. 1f, P<0.05, two-tailed Student's t-test). The difference in YFP-marked clones was even more drastic after a long-term chase (that is, >5 months). At this stage, we found in many areas of MGs from the injected Trp53$^{L/L}$;R26Y females that appeared histologically normal, the ducts were composed of almost entirely YFP$^{+}$ luminal cells (Fig. 1e, right). In these mutant MGs, we also observed extensive alveolar differentiation and the alveoli were composed of almost entirely YFP$^{+}$ cells (Fig. 1e, right, yellow arrows). Quantification of YFP$^{+}$ clones further revealed that

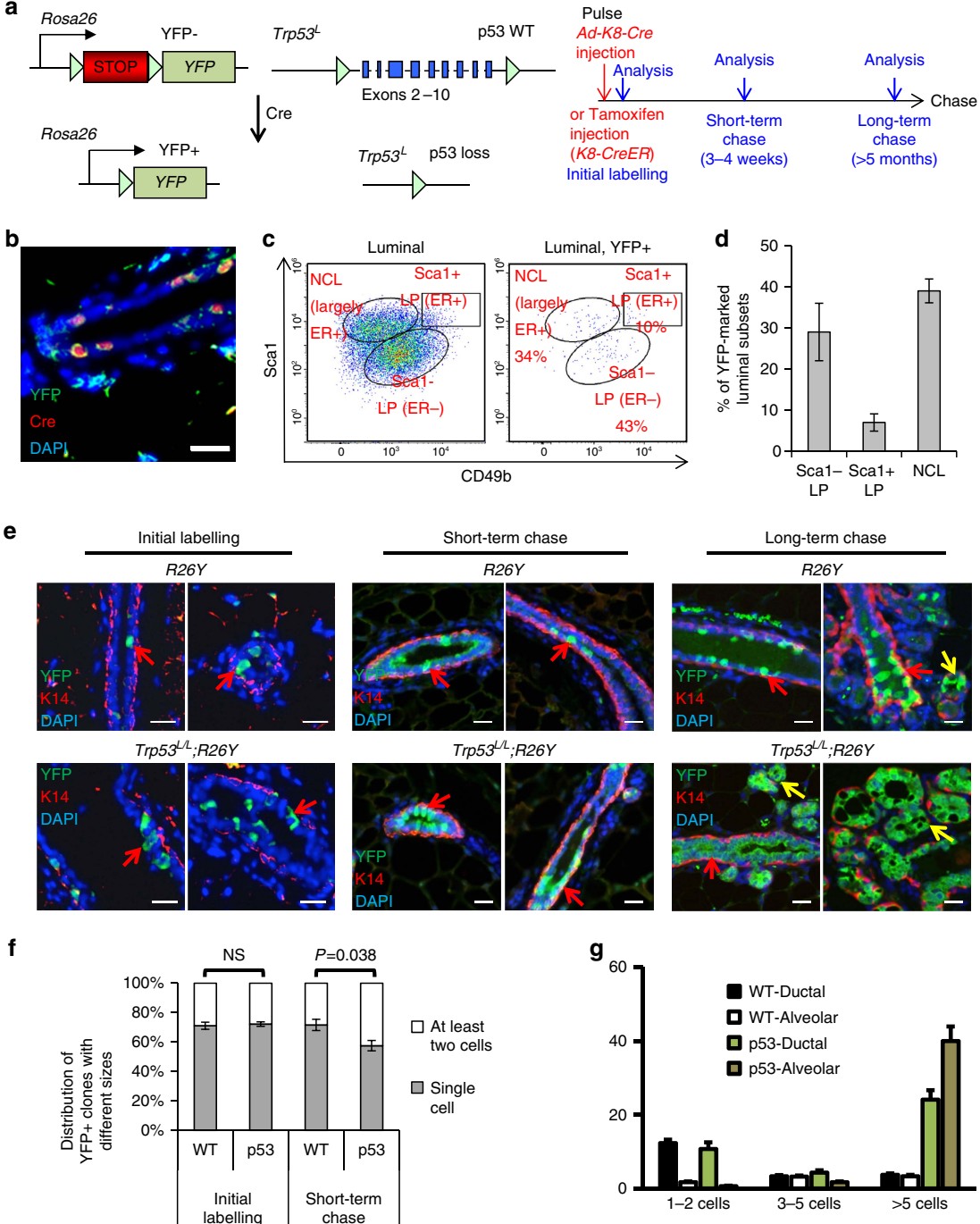

**Figure 1 | Induced loss of p53 in luminal cells leads to their clonal expansion.** (**a**) Schematic diagrams of lineage tracing: Cre (either from transient expression from intraductally injected *Ad-K8-Cre* or upon tamoxifen-induced activation of *K8-CreER*)-mediated excision of either a floxed transcriptional *Stopper* cassette (STOP) in the *R26Y* reporter, or floxed exons 2–10 in the *Trp53^L* allele, leads to YFP expression and disruption of *Trp53* simultaneously (Pulse). The induced females were analysed on chasing for various time periods. (**b**) Representative co-IF picture showing YFP (green) and Cre (red) staining on a MG section from a *Trp53^L/L^;R26Y* female 3 days after *Ad-K8-Cre* injection. *Trp53^L/L^;R26Y* females (*n* = 3); *R26Y* females (*n* = 3). Scale bar, 20 μm. (**c**) FACS analysis showing YFP-marked NCL cells, Sca1+ ER+ LPs and Sca1− ER− LPs, 3 days after injection of *Ad-K8-Cre* to *R26Y* MGs. (**d**) Quantification for the three luminal subpopulations shown in **c** (*n* = 3). (**e**) Co-IF staining showing presence of YFP-marked MECs (green cells, red arrows) 3 days after *Ad-K8-Cre* injection (representing the initial labelling pattern, left), or on either a short-term (middle) or long-term chase (right). Staining for the basal marker Keratin 14 (K14) was used to indicate basal MECs (red cells). On a long-term chase, examples of extensive expansion of YFP+ *Trp53*-null ductal luminal cells (lower right panels, red arrow) and alveolar luminal cells (lower right panels, yellow arrows) were shown; corresponding YFP-marked ductal (upper right panels, red arrows) and alveolar (upper right panels, yellow arrow) luminal cells from WT control mice were also shown. Scale bars, 20 μm. (**f**) Quantification of percentages of YFP-marked single-cell and multi-cell clones in MGs of *Trp53^L/L^;R26Y* (p53) or *R26Y*-only females (WT) as shown in **e** (initial labelling: *n* = 4 for WT, *n* = 3 for p53; short-term chase: *n* = 7 for WT, *n* = 4 for p53). *P* value: NS, not significant, two-tailed Student's *t*-test. (**g**) Quantification of the numbers and types of YFP-marked clones in MGs of the injected *Trp53^L/L^;R26Y* (p53, *n* = 5) or *R26Y*-only females (WT, *n* = 4) after long-term chase, as shown in **e**. In both **f**,**g** at least 10 ducts per section were counted. Data represent mean ± s.e.m.

induced loss of p53 led to the formation of many more large ductal and alveolar clones than WT controls (>5 cells, Fig. 1g).

Thus, our lineage-tracing and clonal analysis data suggested that induced loss of p53 in the luminal lineage might lead to expansion of *Trp53*-null luminal MECs. To further confirm this, we measured percentages of YFP-marked MECs by FACS ~3–4 weeks after *Ad-K8-Cre* injection (Fig. 2a) or after tamoxifen induction (in *K8-CreER;Trp53$^{L/L}$;R26Y* females, Fig. 2b). We found that the percentages of YFP$^+$ populations in p53 experimental females were often increased compared with those of matched WT controls (~two-fold, Fig. 2a–c).

As in our experimental system, YFP$^+$ cells are *Trp53*-null, whereas their YFP$^-$ neighbour cells are *Trp53* WT, our observations suggest that in mammary ducts, induced loss of p53 in luminal MECs may lead to clonal expansion of *Trp53*-null (YFP$^+$) luminal cells, due to their net growth advantage over their *Trp53*-WT neighbours. In addition, induced p53 loss in luminal MECs also leads to aberrant alveolar cell expansion in virgin mice. Together, these data suggest that in addition to increasing the transplantability of MECs in general[29], and the clonogenic activity of both basal and luminal stem/progenitor cells *in vitro*[27], loss of p53 in the luminal lineage leads to clonal expansion of both ductal and alveolar luminal cells in MGs, under the physiological setting.

**Loss of p53 in luminal cells does not affect luminal fate**. Although induced loss of p53 in the luminal lineage led to clonal expansion of *Trp53*-null luminal cells, by co-IF staining of the above MG sections with keratin markers, we found that the expanded YFP$^+$ MECs remained as K8$^+$K14$^-$ luminal cells, even after the long-term chase (Figs 1e and 2d). To confirm this observation, we also performed FACS analysis, by separating luminal and basal MECs based on staining of CD24 and CD29 or CD49f (refs 13,14). By FACS, we found that 3–4 weeks after induced p53 loss, YFP-marked MECs remained within the Lin$^-$CD24$^{hi}$CD29(or CD49f)$^{lo}$ luminal gate (Fig. 2a,b,e). These data suggest that induced loss of p53 alone in luminal MECs does not affect their luminal identity directly, but leads to their clonal expansion.

**Loss of p53 in luminal cells increases their proliferation**. To gain insights into how induced p53-loss affected the luminal lineage at the molecular level, we sorted matched YFP$^+$ MECs from *K8-CreER;Trp53$^{L/L}$;R26Y* females and *K8-CreER;R26Y* control females 4 weeks after tamoxifen injection and subjected them to microarray expression profiling. By gene set enrichment analysis (GSEA)[30], we found that many gene sets related to cell cycle activity were enriched in *Trp53*-null MECs (Fig. 3a), whereas those related to hypoxia, p53 pathway and apoptosis, were enriched in WT MECs (thus downregulated in *Trp53*-null MECs; Fig. 3b and Supplementary Fig. 1a). To confirm this observation, we measured proliferation and apoptosis of YFP$^+$ MECs 3–4 weeks after induced p53 loss. We found that among YFP-marked MECs, *Trp53*-null YFP$^+$ MECs contained more Ki67$^+$ cells than those of WT controls (Fig. 3c and Supplementary Fig. 1b), suggesting expansion of *Trp53*-null luminal MECs was in part due to their increased proliferation and cell cycle activity.

**Immune genes are downregulated in *Trp53*-null luminal MECs**. Intriguingly, among gene sets downregulated in *Trp53*-null MECs, the most significant ones were those related to chemo-kines/cytokines and inflammatory/immune response (Fig. 3b, nominal *P* values <0.001, based on GSEA[30]). Among these, the most notable downregulated genes were chemokine genes

(for example, *Ccl2*, *Ccl3*, *Cxcl2*; Fig. 3d and Supplementary Fig. 1c). CCL2 is a key MEC-secreted chemokine that mediates macrophage recruitment, phagocytosis and activation in the MG[31]. As key immune cells in the MG microenvironment, macrophages play dual roles in ovarian cycle-associated development and remodelling of MECs[32], by promoting proliferation (and differentiation) of luminal MECs following ovulation, and by clearing apoptotic MECs to remodel the mammary epithelium back to its basic architecture for the next oestrous cycle. Macrophages have at least two activation states (that is, polarization states[33]): the classically activated M1 state, in which macrophages are proinflammatory and are cytotoxic against microbes and tumour cells; the alternatively activated M2 state, in which macrophages play a role in homeostatic mechanisms that terminate inflammatory responses and promote wound healing and tissue remodelling[33–35]. Among MEC-expressed cytokines, CSF1 is a key cytokine for recruitment, survival, proliferation and differentiation of most macrophages; in contrast, TGFB1 is involved in M2 macrophage polarization and in repressing the M1 state, whereas CCL2 is proinflammatory[31]. In *Trp53*-null luminal MECs, we found that while *Csf1* expression was unaltered, *Tgfb1* expression was increased, and *Ccl2* expression was reduced (Supplementary Fig. 1d). This expression pattern, combined with downregulation of the inflammatory/immune response-related signatures (Fig. 3b), suggest that induced loss of p53 in luminal MECs may orchestrate an immunosuppressive local microenvironment by inducing an M2-like activation state of their nearby macro-phages and/or by causing reduced macrophage recruitment. The exact nature of this immune microenvironment-related mechanism awaits further investigation.

**Loss of p53 leads to expansion of ER$^+$ ductal luminal cells**. To further determine whether there is any lineage change in *Trp53*-null MECs, we analysed the above-described expression data by using a previously generated collection of MEC subset-specific gene sets[36]. By GSEA, we found that the MaSC (basal MEC) signature was slightly downregulated (rather than upregulated) in *Trp53*-null MECs compared with WT controls (Supplementary Fig. 1e), thus confirming that induced loss of p53 in the luminal lineage does not lead to a luminal-to-basal cell fate change. In addition, we also observed significant upregulation of several ER$^+$ ML and FOXA1 (also known as HNF3A, involved in ER signalling)-related gene sets in *Trp53*-null MECs (Fig. 3a,e; nominal *P* values <0.005, based on GSEA[30]), thus further suggesting maturation of luminal MECs is not impaired. Since we showed that *K8*-based genetic marking could target many ER$^+$ luminal cells (Fig. 1c,d), we reasoned that enrichment of the ER$^+$ ML signature might also suggest that ER$^+$ luminal MECs undergo expansion upon induced loss of p53. To test this, we performed co-IF staining of MG sections 3–4 weeks after injection of *Ad-K8-Cre*. We found that in both control and p53 experimental females, >50% of YFP-marked MECs stained positive for ERα; in control MGs, we found that YFP$^+$ ERα$^+$ MECs typically represented single-cell clones, whereas in p53-mutant MGs, we often observed YFP$^+$ multi-cell clones containing multiple ERα$^+$ MECs (Fig. 3f). This observation thus confirmed that induced loss of p53 in the luminal lineage triggers proliferation of ER$^+$ ductal luminal cells.

At the individual gene level, we found that while many proliferation-related genes were upregulated, p53 target genes related to cell cycle and apoptosis control, such as *Cdkn1a (p21)*, *Bax*, *Bid* and *Fas*, were slightly downregulated in *Trp53*-null luminal MECs (Fig. 3d). We also observed slight upregulation of ER$^+$ ML-related genes (for example, *Esr1*, *Foxa1*, *Prom1*) and

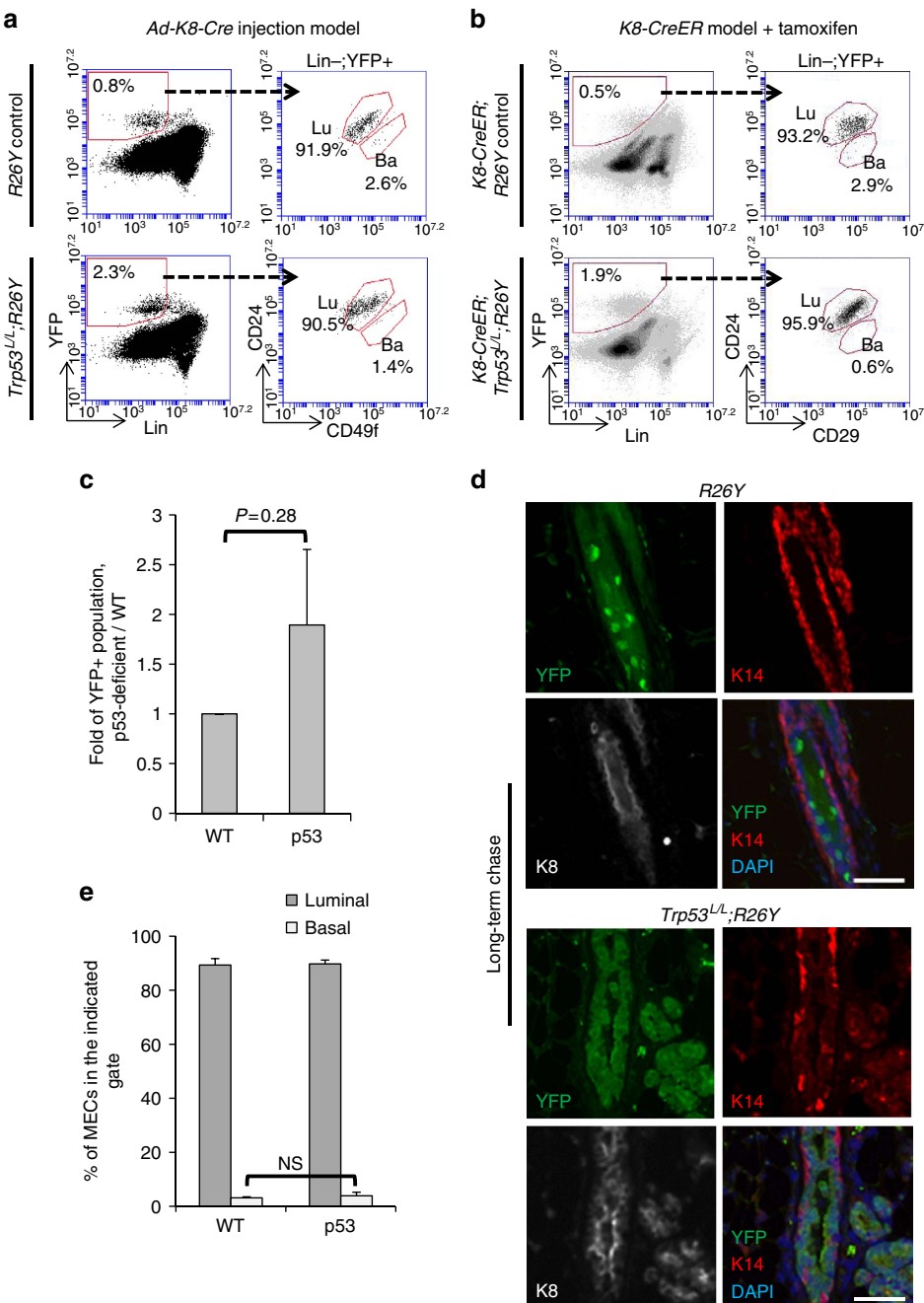

**Figure 2 | Induced loss of p53 in luminal cells does not directly alter their luminal identity.** (**a**) FACS analysis of MGs from virgin females with the indicated genotypes 3 weeks after intraductal injection of *Ad-K8-Cre* showing YFP-marked MECs were largely restricted to the luminal gate. Ba, basal gate; Lin, lineage markers (including CD45, CD31 and Ter119); Lu, luminal gate. (**b**) FACS analysis showing YFP-marked MECs from *K8-CreER;Trp53^{L/L};R26Y* (*n* = 4) or *K8-CreER;R26Y* (*n* = 3) females 4 weeks after tamoxifen administration. Note in both types of mice, YFP⁺ MECs were restricted to the luminal gate. Ba, basal gate; Lu, luminal gate. (**c**) Quantification of percentages of YFP⁺ populations 3–4 weeks after injection of *Ad-K8-Cre* to *Trp53^{L/L};R26Y* females (p53) or matched *R26Y*-only females (WT). Data represent four independent experiments. In each experiment (one WT, one p53), the percentage of YFP⁺ cells in the p53 experimental female was normalized to that in the WT control female ( = 1) from the same experiment. (**d**) Co-IF staining of K8 (white), K14 (red) and YFP (green) on MG sections from *Trp53^{L/L};R26Y* (*n* = 5) or *R26Y*-only (*n* = 4) virgin females > 5 months after intradcutal injection of *Ad-K8-Cre*. Each channel of the co-IF staining is shown. Scale bars, 50 μm. (**e**) Quantification of percentages of cells in the luminal or basal gate in **a**. Data represent mean ± s.e.m. from three independent experiments (one WT, one p53 in each experiment). *P* value: NS, not significant, two-tailed Student's *t*-test. Data represent mean ± s.e.m.

very profound downregulation of multiple chemokine/cytokine genes in *Trp53*-null luminal MECs (Fig. 3d and Supplementary Fig. 1f). We did not observe consistent changes in expression of genes in other categories, such as those related to epithelial-to-mesenchymal transition (EMT; Fig. 3d). Collectively, our data

suggest that induced loss of p53 in luminal MECs does not block luminal differentiation or lead to their dedifferentiation to multipotent basal MaSC-like cells (or lead to aberrant basal differentiation), but causes increased proliferation of *Trp53*-null luminal MECs, in particular ER⁺ ductal luminal cells.

**Loss of p53 in luminal MECs leads to mammary tumours.** We followed these mice for several months upon induced p53 loss in luminal cells and interestingly, starting from ∼6 to 7 months after induction, all female mice (maintained as nulliparous females) developed mammary tumours (Fig. 4a). At this stage, in addition to the histologically normal large patches of *Trp53*-null luminal MECs (Fig. 1e, long-term chase), we also detected histologically abnormal *Trp53*-null premalignant lesions

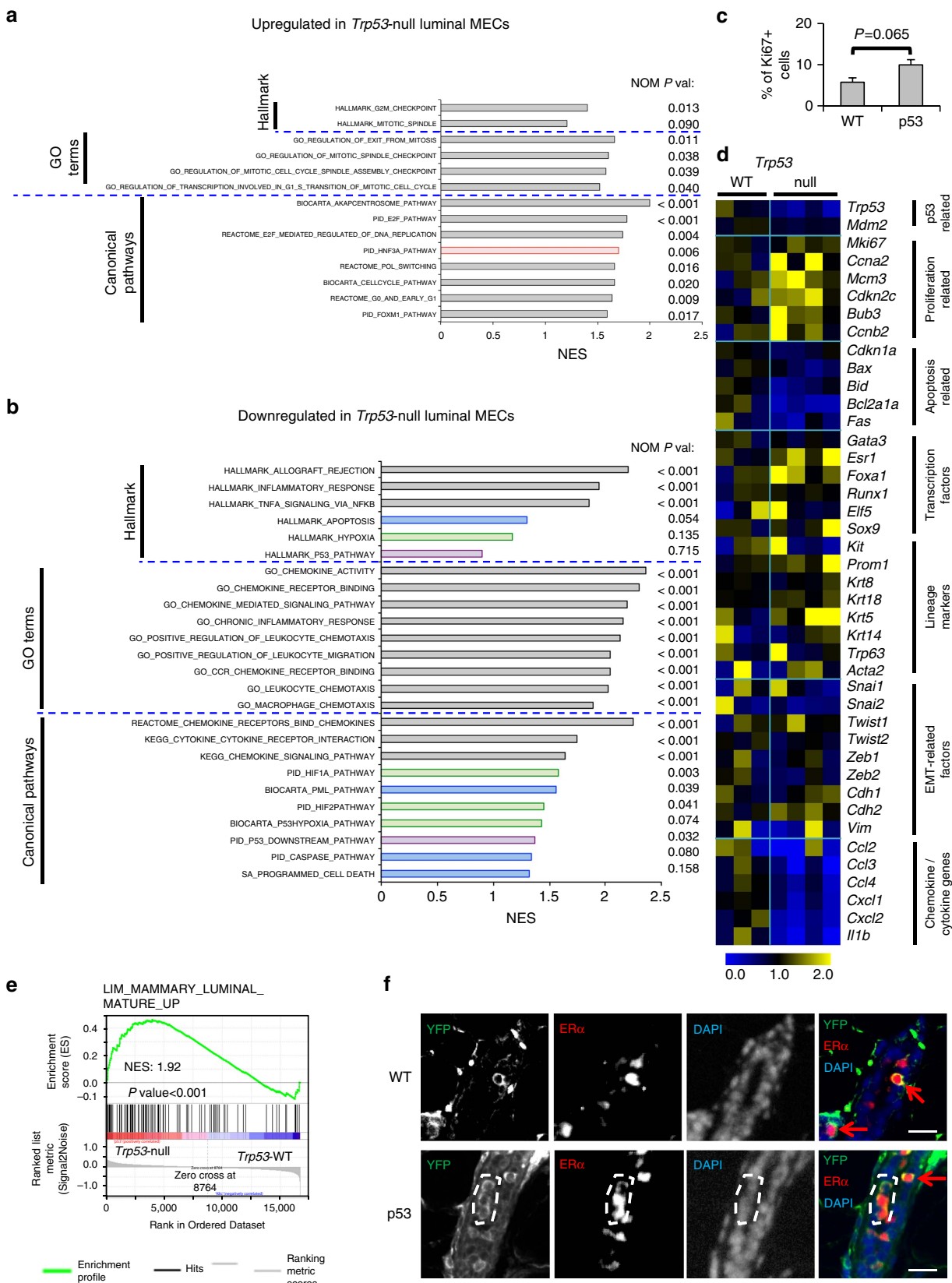

(Supplementary Fig. 2a–c). Of note, all these aberrant pre-malignant lesions contained MECs with luminal-to-basal change (that is, aberrant $K8^+K14^+$ or $K8^+K5^+$ cells, Supplementary Fig. 2a–c, arrows). We characterized mammary tumours developed in these induced females and found that the majority of them ($\sim80\%$) were large, fast growing carcinosarcomas that exhibited a mesenchymal cell appearance (Type 1 tumours, Fig. 4b and Supplementary Fig. 2d,e). Most of them were $YFP^+$ and were composed of many $K8^-K14^-$ mesenchymal-like tumour cells. Consistent with this, by staining for a Claudin marker, Claudin 3, as well as an epithelial marker E-Cadherin, we found that these tumours were negative for both (Fig. 4b), suggesting a loss of their epithelial identity. Nevertheless, in the majority of these tumours, we could still detect keratin-expressing epithelial tumour cells at varying degrees, including K8 and K14 single and/or double positive cells, and to a lesser degree, $K5^+$ cells (Fig. 4b and Supplementary Fig. 2e). In addition to these carcinosarcomas, we also observed a small number of mammary tumours ($\sim20\%$) that were smaller in size and were mainly composed of $K8^+K14^+$ basal-like epithelial cells. These tumours also expressed Claudin 3 and E-Cadherin, but contained fewer $K5^+$ cells (Type 2 tumours, Fig. 4c and Supplementary Fig. 2d,f). Keratin staining patterns of all Type 1 and 2 tumours are summarized in Supplementary Table 1. We subjected some of these tumours to microarray expression profiling. By GSEA, we compared their expression profiles with expression signatures of human and murine breast cancer intrinsic subtypes (based on ref. 37) individually. Their similarities to human and murine intrinsic subtypes were assessed by GSEA enrichment scores and $P$ values (Supplementary Data 1). We found that all these tumours most closely resembled the human and murine claudin-low subtypes at the molecular level, followed by the human basal-like subtype and murine $C3Tag^{Ex}$, $Myc^{Ex}$ and $p53null\text{-}Basal^{Ex}$ subtypes (these three murine subtypes represent murine models for $BLBC^{37}$); they also exhibited a lower level of similarity to the human luminal B subtype and murine $p53null\text{-}Luminal^{Ex}$ subtype (Fig. 4d,e). Of note, among these profiled tumours, the Type 2 tumour E8 exhibited the lowest similarity to the human and murine claudin-low subtypes but the highest similarity to the human basal-like subtype (Fig. 4d,e). Furthermore, compared with WT and *Trp53*-null luminal MECs (that is, from mice 4 weeks after induction), all these tumours, in particular the Type 1 tumours, exhibited lower expression levels of multiple Claudin genes (for example, *Cldn3*, *Cldn4*, *Cldn7*), and higher expression levels of many EMT-related genes (for example, *Twist1*, *Zeb1*, *Cdh2* and *Vim*; Fig. 4f). Along the MEC differentiation hierarchy, all tumours exhibited significant upregulation of the MaSC signature (Fig. 4g; nominal $P$ value $<0.001$, based on $GSEA^{30}$) and downregulation of luminal signatures (Supplementary Fig. 2g; nominal $P$ values $<0.001$,

based on $GSEA^{30}$). Overall, our observations from both premalignant lesions and mammary tumours in these induced mice suggest that tumours developed in them most closely resemble the claudin-low subtype, and to a lessor degree, the basal-like subtype; common features of all these tumours include their loss of luminal identity, gain of basal-like and mesenchymal-like properties (at varying degrees), and acquisition of a MaSC-like molecular signature.

**Recurrent genomic lesions in *Trp53*-null mammary tumours.** Since all tumours developed in these animals have a luminal origin and induced loss of p53 in luminal MECs does not lead to loss of the luminal cell fate directly, we reasoned that other oncogenic events, such as those caused by genomic abnormalities triggered by p53 loss, might drive *Trp53*-null luminal MECs toward a basal or mesenchymal-like cell fate. To determine this, we performed high-density single-nucleotide polymorphism (SNP) array analyses of genomic DNAs from four Type 1 tumours (E2, E4, E5, E9) and one Type 2 tumour (E8), as well as from MGs of a genetic background-matched WT female. By comparing with signals from WT, we found that all Type 1 tumours only carried one common genomic lesion, an over-lapping 1.15 Mb amplicon on chromosome 9 (9qA1) that contains multiple Matrix metalloproteinase (Mmp) genes, as well as *Yap1*, which encodes a key downstream nuclear effector of the Hippo pathway[38], and *Birc2* and *Birc3*, which encode IAP family of proteins that inhibit apoptosis (Fig. 5a, left). Supporting amplification of this region, analysis of our microarray data revealed much higher expression levels of multiple genes (for example, *Dync2h1*, *Mmp3*, *Mmp10*, *Birc2*, *Birc3*, *Yap1*) from this region in almost all profiled Type 1 tumours (for example, E2, E4, E5, E9, TB208, TB209, TB239-2), but not in the Type 2 tumour E8 (Fig. 5b, top). Of note, amplicons encompassing *Yap1* or *Birc2*, *Birc3* and *Yap1* were also found previously in mammary tumours developed in other breast cancer mouse models with p53 deficiency[39,40]. Importantly, overexpression of YAP1 in human MCF10A breast cells led to EMT, inhibition of apoptosis and proliferative advantage in the transduced cells[39]. These observations suggest that elevated expression of *Yap1* may cooperate with p53 loss to drive development of claudin-low mammary tumours from luminal cells; meanwhile overexpression of *Mmp* genes and anti-apoptotic genes *Birc2/3* may also contribute to this process. Supporting a potential cooperation between YAP1 overexpression and p53-loss, we found that higher YAP1 expression is only associated with a worse outcome in human breast cancer cases with *TP53* mutations (marginally), but not in those with WT *TP53* or in BLBCs (Supplementary Fig. 3). In addition to the 9qA1 amplicon, some Type 1 tumours (E4, E5) also carry a common amplicon on chromosome 3 (Fig. 5c).

**Figure 3 | Microarray analysis and validation of luminal MECs on induced loss of p53.** (**a**) GSEA results showing upregulation of proliferation-related (grey-shaded) and FOXA1 (HNF3A)-related (red-shaded) gene sets from the H (Hallmark), C2 (canonical pathways) and C5 (Gene Ontology (GO) terms) collections of the GSEA MSigDB v5.2 in $YFP^+$ *Trp53*-null MECs from *K8-CreER;Trp53^{L/L};R26Y* females ($n=4$) compared with $YFP^+$ WT MECs from *K8-CreER;R26Y* females ($n=3$), 4 weeks after tamoxifen induction. (**b**) GSEA results showing downregulation of immune (grey-shaded), hypoxia (green-shaded), p53 pathway/targets (purple-shaded) and apoptosis (blue-shaded)-related gene sets in $YFP^+$ *Trp53*-null MECs. In **a,b** normalized enrichment score (NES) and nominal $P$ value (NOM p-val) are shown for each gene set. (**c**) Percentages of $Ki67^+$ cells among YFP-marked MECs in *Trp53^{L/L};R26Y* females (p53, $n=3$) or *R26Y*-only females (WT, $n=3$) 3–4 weeks after *Ad-K8-Cre* injection. Representative YFP/Ki67 co-staining pictures are shown in Supplementary Fig. 1b. At least 10 ducts per section were counted. Data represent mean ± s.e.m. (**d**) Heatmap showing expression levels of select genes in different categories in $YFP^+$ *Trp53*-null MECs compared with matched $YFP^+$ WT MECs. For each gene, the expression value from each sample was normalized to the mean of WT samples ($=1$) and the resulted relative expression value (that is, fold change to the mean of WT) was shown on the heatmap. (**e**) GSEA result showing significant enrichment of an $ER^+$ mature luminal cell gene set (from ref. 36) in *Trp53*-null MECs in relation to WT controls. NES and nominal $P$ value are shown. (**f**) Representative co-IF pictures showing YFP (green) and ERα (red) staining for MGs from *Trp53^{L/L};R26Y* females (p53, $n=4$) or *R26Y*-only females (WT, $n=4$) 3–4 weeks after *Ad-K8-Cre* injection. An $YFP^+$ multi-cell clone comprised of two $ER^+$ cells and one $ER^-$ cells is highlighted. Individual $YFP^+ER^+$ cells are indicated by red arrows. Scale bars, 20 μm.

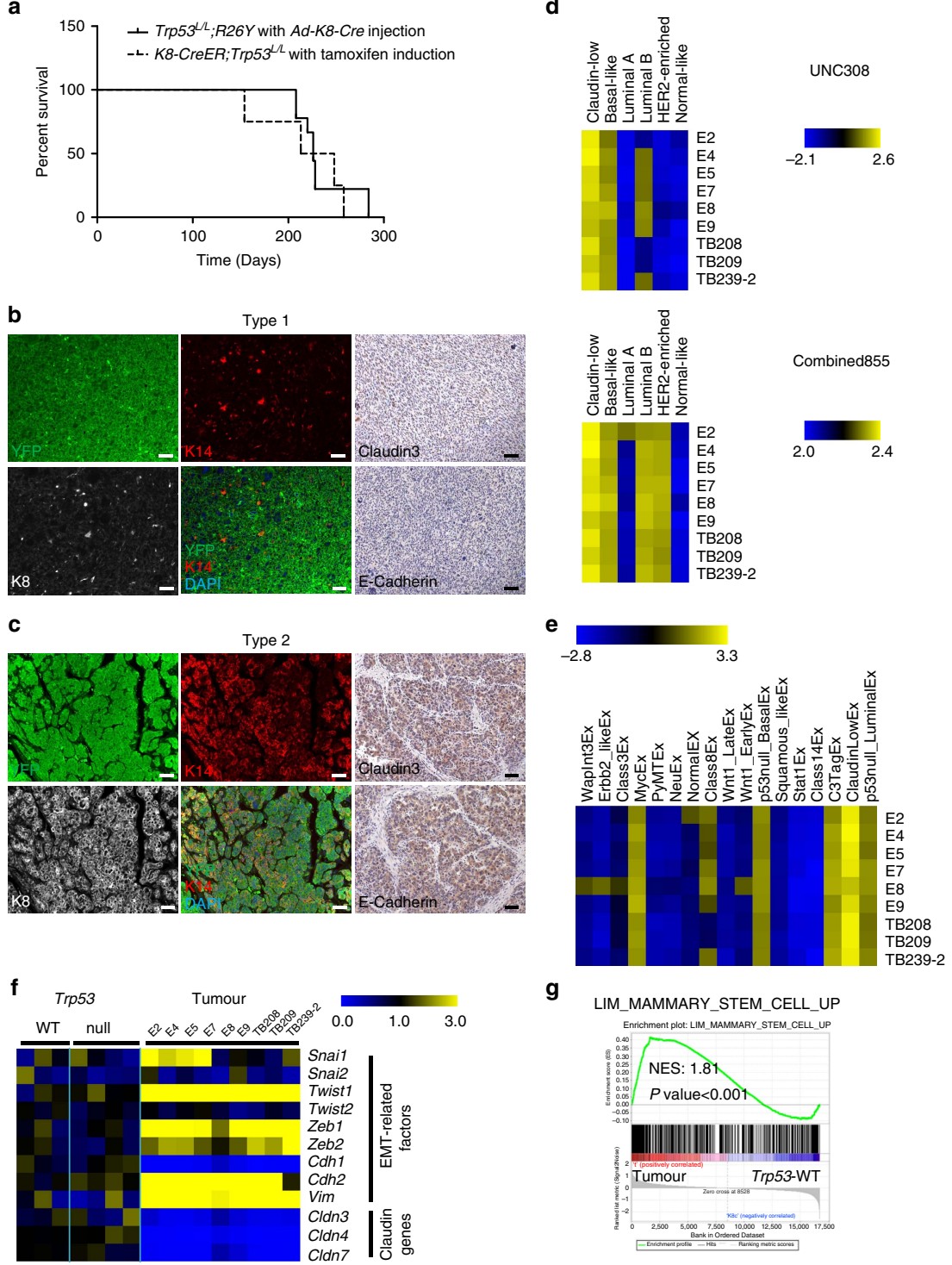

**Figure 4 | Induced loss of p53 in luminal cells leads to development of mammary tumours with loss of luminal identity. (a)** Kaplan–Meier survival curves showing all *Trp53*$^{L/L}$ females (*n* = 9) injected with *Ad-K8-Cre* or *K8-CreER;Trp53*$^{L/L}$ females (*n* = 4) induced by tamoxifen (2 mg per mouse, one injection) developed mammary tumours with a similar latency. **(b,c)** Co-IF staining (for YFP, K8, K14, left and middle panels) and immunohistochemical (IHC) staining (for Claudin 3 and E-Cadherin, right panels) showing that most mammary tumours developed in cohorts in **a** were YFP$^+$ mammary tumours that were largely negative for K8 and K14, and were negative for Claudin 3 and E-Cadherin (**b**, Type 1). In addition, a small number of mammary tumours developed in cohorts in **a** were YFP$^+$;K8$^+$;K14$^+$;Claudin 3$^+$;E-Cadherin$^+$ mammary tumours (**c**, Type 2). Scale bars, 50 μm. **(d,e)** Heatmaps summarizing GSEA results (plotted as NES, from Supplementary Data 1), based on comparisons of microarray data of each tumour to those of *Trp53*-WT luminal MECs and to gene sets representing human (**d**) and murine (**e**) intrinsic subtypes. The gene sets were extracted from Pfefferle *et al.* (Human intrinsic subtypes were based on the UNC308 data set and Combined855 data set)[37]. **(f)** Heatmap showing upregulation of EMT-related genes and downregulation of E-Cadherin gene (*Cdh1*) and Claudin genes in Type 1 (and to a lessor gene, in Type 2 (E8)) tumours. For each gene, the expression value from each sample was normalized to the mean of WT samples (= 1) and the resulting relative expression value (that is, fold change to the mean of WT) was shown on the heatmap. **(g)** GSEA plot showing significant enrichment of a MaSC-related gene set based on Lim *et al*[36] in all tumours in relation to *Trp53*-WT luminal MECs (that is, cellular origin). NES and nominal *P* value are shown.

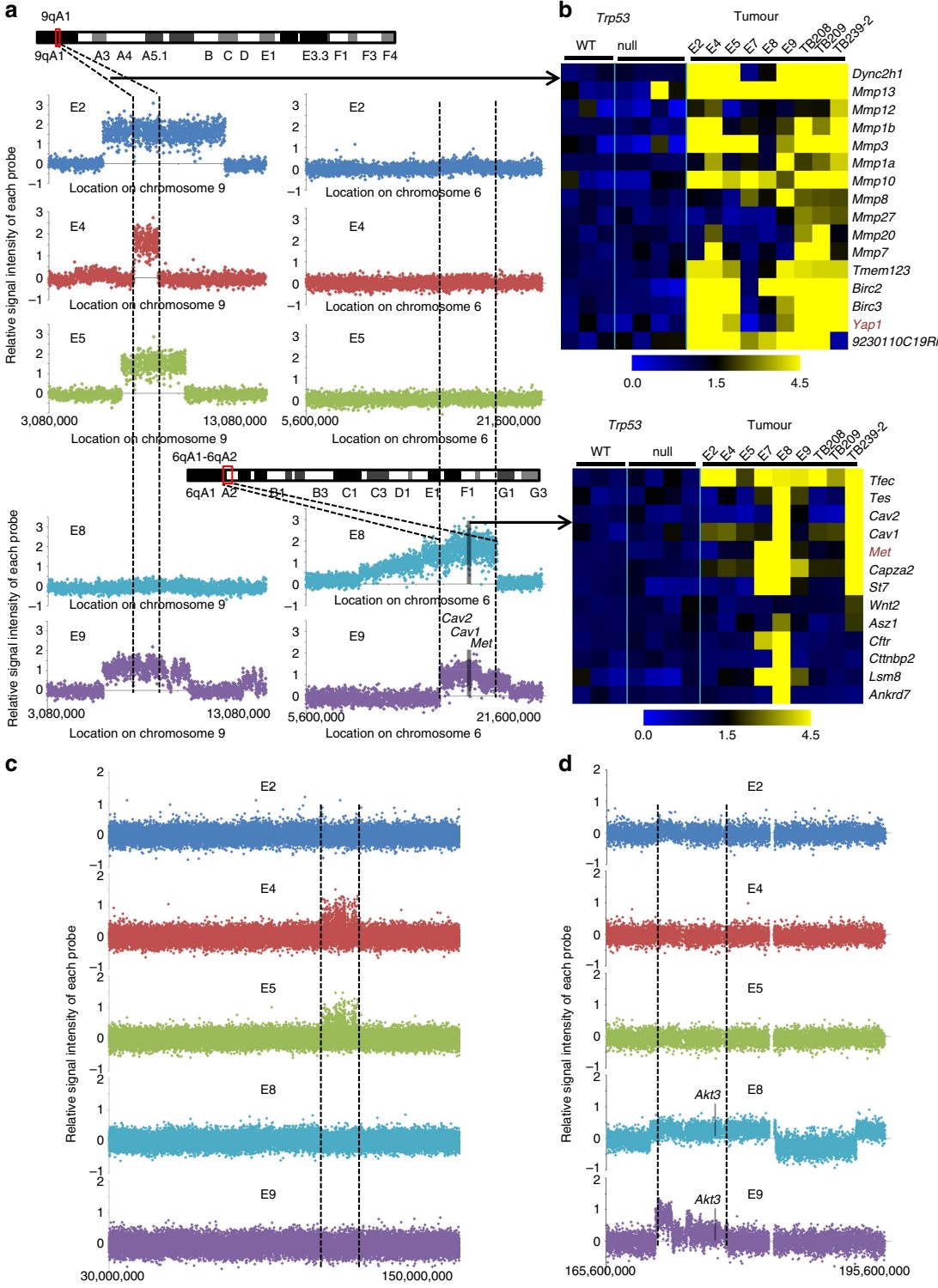

**Figure 5 | SNP array analysis showing recurrent genomic lesions in Type 1 and 2 mammary tumours. (a)** Left: a recurrent genomic lesion (∼1.15 Mb amplicon) on chromosome 9qA1 was identified in several Type 1 tumours (E2, E4, E5, E9). Right: a recurrent genomic lesion (amplicon spanning chromosome 6qA1-6qA2) encompassing *Cav2*, *Cav1* and *Met* was identified in tumours E8 and E9. In all plots, relative signal intensity of each probe is shown as the strength of each probe from each tumour sample minus that from the WT control MGs. **(b)** Top: the recurrent 1.15Mb amplicon on chromosome 9qA1 encompasses multiple *Mmp* and *Birc* genes, as well as the Hippo pathway-related *Yap1* gene; heatmap from microarray confirmed high expression levels of multiple genes (for example, *Yap1*) in this amplicon in tumours E2, E4, E5 and E9 (as in **a**); it also revealed several additional tumours (TB208, TB209, TB239-2) with potential amplification of this region. Bottom: the recurrent amplicon on chromosome 6qA1-6qA2 encompasses *Cav2*, *Cav1* and *Met*; heatmap from microarray confirmed high expression levels of multiple genes (for example, *Met*) in this amplicon in tumours E8 and E9 (as in **a**); it also revealed several additional tumours (E7, TB239-2) with potential amplification of this region. In both heatmaps, expression values were normalized to mean of WT samples (=1). **(c)** A common amplicon on chromosome 3 identified only in a subset of Type 1 tumours (E4, E5). **(d)** A common amplicon on chromosome 1 identified only in tumours E8 and E9. A potential BLBC-related oncogene *Akt3* in this amplicon is indicated.

In addition to 9qA1 (*Yap1*) amplification, in tumours E8 and E9, we also detected genomic abnormalities on chromosomes 1 and 6. On chromosome 6, a common amplicon encompassing 6qA1-A2 contains *Cav2*, *Cav1* and *Met* (Fig. 5a, right). *Met* is a well-known oncogene that drives development of BLBC[41,42]; however, it also drives development of claudin-low mammary tumours when under the p53-loss background[43]. *Cav1* and *Cav2* are two caveolin genes associated with human BLBC[44]. Analysis of microarray data revealed higher expression levels of *Met* in E8 as well as in several Type 1 tumours (for example, E7, TB239-2; Fig. 5b, bottom), suggesting *Met* amplification might also be a relatively common event in these tumours. Lastly, on chromosome 1, both tumours E8 and E9 have a common amplicon at 1qH3-H4 encompassing *Akt3* (Fig. 5d); of note, high expression of *AKT3* is a feature of human BLBC[2].

Next, we expanded our analysis to additional tumours from these mice, focusing on expression changes and potential genomic amplifications of the *Yap1*, *Met* and *Akt3* genes. Consistent with development of predominantly claudin-low carcinosarcomas in these animals, we found almost all these tumours, except two (E7, E8), exhibited elevated *Yap1* expression, which were accompanied by *Yap1* gene amplification (Fig. 6a). In addition, one-third of all analysed tumours, including E7 and E8, exhibited notably higher *Met* expression, and almost all of them had genomic amplification at the *Met* locus (except TB242, which exhibited elevated *Met* expression but without *Met* gene amplification; Fig. 6b). Lastly, we found that only tumours E8 and E9 exhibited both elevated *Akt3* expression and amplification of the *Akt3* gene, whereas tumours E1, TB239-2 and TB242 had elevated *Akt3* expression but without *Akt3* gene amplification (Fig. 6c). Overall, our combined SNP array and PCR analyses suggest that 9qA1 (*Yap1*) amplification and 6qA2 (*Met*) amplification are the recurrent genomic lesions that may drive development of mammary tumours (with loss of luminal identity) from *Trp53*-null luminal MECs.

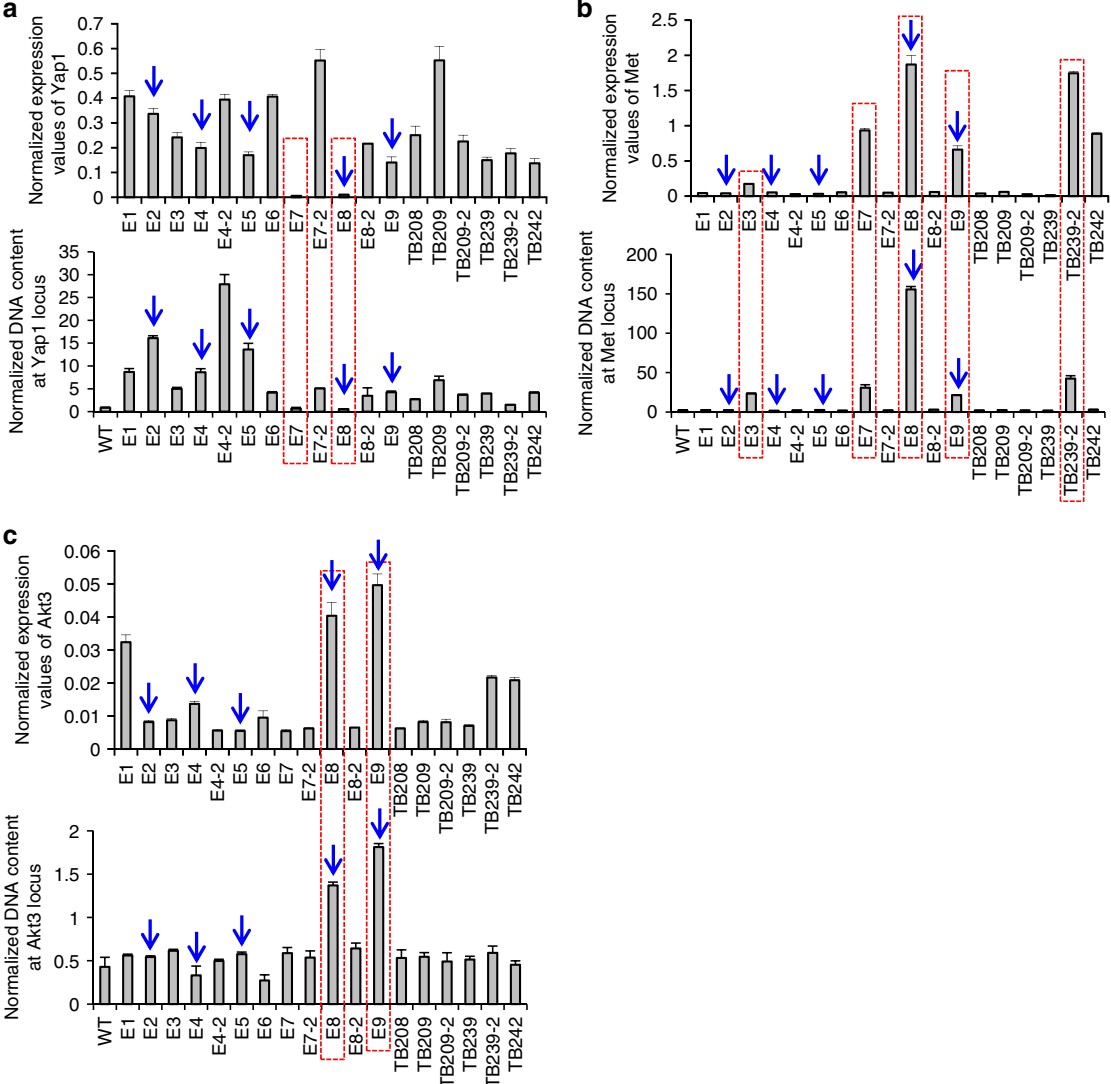

**Figure 6 | PCR analyses confirming recurrent genomic lesions in Type 1 and 2 mammary tumours.** (**a–c**) qRT-PCR (top, normalized to expression levels of *Gapdh*) and genomic PCR (bottom, normalized to the *Lsd1* gene body, which exhibits no amplification based on SNP analysis) analyses showing excellent correlation of *Yap1* (**a**) and *Met* (**b**) overexpression with amplification of their corresponding genes in multiple tumours developed in the *Ad-K8-Cre/Trp53*[L/L] and *K8-CreER;Trp53*[L/L] (tamoxifen) models. In **a**, two tumours (E7, E8) with no *Yap1* overexpression/amplification are highlighted in red. In **b**, five tumours (E3, E7, E8, E9 and TB239-2) with both *Met* overexpression and gene amplification are highlighted in red. The PCR analyses also show correlation of *Akt3* overexpression with its gene amplification in tumours E8 and E9 (highlighted in red) (**c**). In **a–c**, five tumours analysed in Fig. 5a are indicated by arrows. Data represent mean ± s.e.m.

## Discussion

In this study, we showed that induced loss of p53 in the mammary luminal lineage affected both ER$^+$ ductal and ER$^-$ alveolar luminal cells. Expansion of *Trp53*-null ER$^+$ ductal luminal cells was observed as early as 3 weeks after induced p53 loss, whereas aberrant expansion of *Trp53*-null ER$^-$ alveolar luminal cells (in nulliparous females) became evident after a longer latency (Fig. 1). Although based on *in vivo* transplantation and *in vitro* clonogenic assays, it was initially proposed that MECs are organized in a hierarchy in which basal MaSCs produce LPs (largely ER$^-$ luminal MECs), which then give rise to ER$^+$ MLs and ER$^-$ alveolar luminal cells[45], recent studies based on *in situ* lineage-tracing and cell division kinetics studies suggest that in adult homeostatic MGs, the mammary luminal lineage appears to be sustained by its own lineage-restricted stem and progenitor cells[15]; in particular, the ER$^+$ and ER$^-$ luminal MECs may represent separate luminal sub-lineages largely maintained by their own restricted progenitors[24,46]. Therefore, the MEC phenotypes we observed after different chasing time periods may reflect the effects of p53 loss on different luminal sub-lineages. Of note, for the alveolar hyperplasia phenotype, due to its longer latency, we cannot rule out a possibility of contribution of secondary mutation(s) (for example, due to p53 loss) to the expansion of *Trp53*-null alveolar cells.

At the molecular level, we found that induced loss of p53 in luminal MECs led to increased cell cycle activity and reduced control of apoptosis, as well as profound reduction in expression of chemokine/cytokine genes (Fig. 3). In females, due to fluctuation in levels of ovarian hormones at different stages of oestrous cycle, MECs undergo cyclic changes (that is, in a way similar to lactation followed by involution, but in a smaller scale). During each cycle, luminal cells go through sequential proliferation, differentiation and apoptosis, followed immediately by another wave of proliferation when the next cycle starts[47]. This unique repeated stimulation and withdrawal of ovarian hormones in MGs, in a way analogous to ultraviolet light exposure in skin or colitis in small intestine[48,49], may trigger an imbalance of proliferation versus apoptosis between *Trp53*-null and their neighbouring WT luminal MECs, which can result in a net accumulation of *Trp53*-null luminal MECs in nulliparous mice over time. The immune microenvironment (for example, macrophages) in the MG, which is also under regulation of cyclic ovarian hormones, may play a role in shaping the *Trp53*-null luminal phenotype as well.

Our observation that induced loss of p53 leads to expansion of ER$^+$ ductal luminal cells is of particular interest, as in women with Li–Fraumeni Syndrome (that is, with *TP53* germline mutation), the most prevalent invasive breast tumours developed in them are ER$^+$ (ref. 50). Although we did not observe development of ER$^+$ mammary tumours from this mouse model, such tumour was reported previously in a *Trp53*-null transplantation mouse model[9]. Mammary tumours developed in our *Trp53*-null model with a luminal origin recapitulated two types of mammary tumours developed in this *Trp53*-null transplantation mouse model[9], including the claudin-low subtype, and to a lessor degree, the basal-like subtype. A much limited tumour spectrum in our model can be explained by several possibilities: (1) a difference in cells of origin (that is, both luminal and basal MEC origins in the transplantation model versus luminal origin in this study); (2) a difference in the microenvironment in which cancer originates and progresses (that is, in the transplantation model, all MECs are *Trp53*-null, whereas in this study, mammary tumours originate and progress from *Trp53*-null luminal MECs surrounded by WT luminal and basal MECs); (3) a difference in the strain background (that is, BALB/c background in the transplantation model versus FVB

background in this study): of note, the FVB genetic background in our mice led to a considerable shorter tumour latency (that is, ~6–9 months in our model (Fig. 4a) versus ~9–14 months in the BALB/c transplantation model[9]). Genetic modifiers and/or pre-existing mutations in FVB mice may preferentially drive development of claudin-low tumours from *Trp53*-null luminal cells in our model; alternatively, the FVB background may facilitate acquisition of *Yap1* or *Met* amplification in *Trp53*-null luminal cells, which cooperates with p53-loss, leading to development of claudin-low mammary tumours. The shorter latency (of developing claudin-low tumours) in our model may prohibit development of other mammary tumour subtypes (for example, luminal tumours) that require a longer latency.

Although basal MaSCs have been suggested as the cellular origin of claudin-low breast cancer[20], our study provides evidence to support that claudin-low mammary tumours can also develop from luminal MECs. However, our conclusion is not in disagreement with a basal origin of claudin-low breast cancer, as our data, in particular that from the premalignant lesions (Supplementary Fig. 2a–c), revealed a possible transition of *Trp53*-null luminal cells to a K14$^+$ and/or K5$^+$ basal-like intermediate state first, before emergence of claudin-low tumours. Further supporting this notion, we found that the majority of claudin-low tumours from our induced mice also contained K14$^+$ and/or K5$^+$ basal-like cells (Supplementary Table 1), even though they were originally derived from luminal MECs. The Type 2 tumours were less frequently detected in these induced females and were smaller in size, and exhibited more basal-like epithelial cell appearance; they may either represent a separate and stable tumour subtype or represent an intermediate tumour cell state that may eventually progress to a mesenchymal-like state (that is, claudin-low tumours). Nevertheless, further study using a different luminal promoter (for example, *Wap*[24]) is needed to fully support the conclusion of a luminal origin of claudin-low breast cancer. Lastly, it should be emphasized that the premalignant lesions with basal-like change were observed only after a long-term chase (on induced p53 loss), suggesting secondary mutation(s) (for example, due to p53 loss) might have caused the cell fate alteration and loss of p53 might have opened the 'gate' for this change.

All mammary tumours developed in this inducible mouse model express a MaSC-like signature and exhibit features of EMT (Fig. 4f,g). It has been proposed that cellular reprogramming and transformation may use common pathways and represent variations on similar biological themes[51]. Although p53 does not play a direct role in reprogramming of somatic cells to induced pluripotent stem cells, its loss facilitates this progress[52]. We suggest that p53 plays a similar role during development of TNBCs from committed luminal MECs, by serving as a barrier to block acquisition of MaSC-like properties and EMT by committed luminal cells. Disruption of the p53 activity does not directly lead to loss of the luminal identity but facilitates this process, which may explain why *TP53* inactivation mutations are highly prevalent in TNBCs.

## Methods

**Mouse models.** *Trp53$^L$* (B6.129P2-*Trp53$^{tm1Brn}$*/J, Stock No: 008462), *R26Y* (B6.129X1-*Gt(ROSA)26Sor$^{tm1(EYFP)Cos}$*/J, Stock No: 006148) and *K8-CreER* (STOCK Tg(Krt8-cre/ERT2)17Blpn/J, Stock No: 017947) mice were obtained from The Jackson Laboratory. All mice were backcrossed to the FVB/N background for at least 6 generations. To target luminal MECs in *Trp53$^{L/L}$;R26Y* or *R26Y*-only adult female mice (2 months of age), mice were anaesthetized and *Ad-K8-Cre* adenovirus (diluted in injection medium (DMEM supplemented with 0.1% Bromophenol blue and 0.01 M CaCl$_2$)) were introduced into mammary ducts of their #4 inguinal glands via intraductal injection[21]. To induce CreER activity, tamoxifen (2 mg per mouse, one injection or 5mg per mouse, three injections) was introduced into adult female mice (2 months of age) carrying *K8-CreER* by intraperitoneal injection[15]. After induction, female mice were monitored weekly for

any sign of mammary tumour development. Once tumours were detected via palpation or visual inspection, animals were monitored 2–3 times per week for their tumour growth; tumours were allowed to grow before they reached 10–15% of the body weight of the mouse. All animal experiments were approved by the Institutional Animal Care and Use Committee (IACUC) of Boston Children's Hospital where these mice were housed.

**Mammary gland cell preparation and flow cytometry.** Thoracic and inguinal mammary glands were dissected and minced, and then incubated in digestion medium (DMEM/F12 with 2% Penicillin/Streptomycin, 0.1 mg ml$^{-1}$ Gentamicin, 0.6% Nystatin, 2 mg ml$^{-1}$ Collagenase A, 0.096 mg ml$^{-1}$ Hyaluronidase) at 37 °C with shaking for 1–1.5 h. After digestion, the cells/tissues were treated sequentially with 0.25% trypsin/EDTA (37 °C, 2 min), 5 mg ml$^{-1}$ dispase with DNaseI (0.1 mg ml$^{-1}$, Sigma, St Louis, MO; 37 °C, 5 min), cold red blood cell (RBC) lysis buffer (on ice, 2–3 min). Between each treatment step, cells/tissues were washed with 1 × PBS with 5% FCS. After treatment with the RBC lysis buffer, cells/tissues were filtered through 40 μm cell strainer and washed with 1 × PBS/5% FCS, to obtain single-cell suspension[13]. Flow cytometric (FACS) analysis was performed by using an Accuri C6 analyzer (BD Biosciences, San Jose, CA) and analysed with CFlow software (BD Biosciences). FACS cell sorting was performed with a FACSAria sorter (BD Biosciences). Antibodies used in FACS analysis and cell sorting were purchased from eBiosciences (San Diego, CA) and included, CD24-PE (clone M1/69, 12-0242-83; 1:250), CD29-APC (clone eBioHMb1-1, 12-0291-82; 1:250), CD49f-APC (clone eBioGoH3 (GoH3), 12-5971-81; 1:250), Ly-6A/E (Sca-1) PE-Cyanine7 (clone D7, 25-5981-81; 1:250) and biotinylated CD31 (clone 390, 13-0311-85; 1:100), CD45 (clone 30-F11, 17-0495-82; 1:100) and TER119 (clone Ter-119, 13-5921-85; 1:100) (that is, lineage (Lin) markers), or from BioLegend (San Diego, CA) and included CD49b-AF647 (clone HMα2, 103511; 1:250) and CD24-AF700 (clone M1/69, 564237; 1:250), or from BD Biosciences and included BV605-Streptavidin (563260; 1:250).

**Immunostaining and clonal analysis.** Immunohistochemistry (IHC) and immunofluorescence (IF) staining was performed on sections from inguinal mammary glands or from mammary tumours that were fixed in 10% formalin (Fisher Scientific, Hampton, NH) and embedded in paraffin. Antigen retrieval (Citrate buffer pH 6.0, 20 min boil in microwave oven) was performed before blocking and endogenous peroxidase activity was quenched on the slides intended for IHC by incubation in 0.3% H$_2$O$_2$. Primary antibodies used included: anti-GFP (ab290, 1:500 or ab6673, 1:200, Abcam, Cambridge, UK), anti-Keratin 14 (K14) (PRB-155P, 1:400 or SIG-3476, 1:200, Covance, Dedham, MA), anti-Keratin 5 (K5) (PRB-160P, 1:500, Covance), anti-Keratin 8 (K8) (MMS-162P, 1:200, Covance), anti-Claudin 3 (18-7340, 1:100, Invitrogen, Carlsbad, CA), anti-E-Cadherin (3195, 1:100, Cell Signaling, Danvers, MA), anti-ERα (sc-542, 1:100, Santa Cruz, Dallas, TX), anti-Ki67 (sc-7846, 1:500, Santa Cruz) and anti-Cre (NB100-56133, 1:100, Novus, Littleton, CO). For IHC, signal was detected using the impress reagent kit and DAB substrate (MP-7401 and SK-4100, Vector Laboratories, Burlingame, CA). For IF staining, the secondary antibodies used were goat anti-mouse IgG conjugated with AF488 (A11029, 1:250) or with AF647 (A31571, 1:250), donkey anti-goat IgG conjugated with AF488 (Ab150129, 1:250), chicken anti-goat IgG conjugated with AF647 (A21469, 1:250), goat anti-rabbit IgG conjugated with AF488 (A11008, 1:250) or with AF594 (A11037, 1:250) and goat anti-chicken IgG conjugated with AF594 (A11042, 1:250) or with AF488 (A11039, 1:250) (all from Molecular Probes, Eugene, OR). Slides were counterstained with either hematoxylin (IHC) or DAPI (IF, 1 μg ml$^{-1}$). To determine the clone sizes of YFP-marked cells in MGs of $Trp53^{L/L}$;$R26Y$ or $R26Y$-only females injected with $Ad$-$K8$-$Cre$, YFP$^+$ clones were counted in MG sections from 4 to 7 injected mice for each genotype. Single or clusters of YFP$^+$ cells that contact each other were defined as YFP-marked clones. Clones were further grouped by the number of YFP$^+$ cells per clone and were scored[21,24].

**Microarray expression profiling and analysis.** Total RNAs from YFP$^+$ MECs sorted from $K8$-$CreER$;$Trp53^{L/L}$;$R26Y$ or $K8$-$CreER$;$R26Y$ females 4 weeks after tamoxifen induction or from $Trp53$-null mammary tumours were prepared by the RNeasy kit (Qiagen, Valencia, CA). The NuGen (San Carlos, CA) Ovation Pico WTA System V2 kit was used with 500 pg of starting RNA. Five micrograms of DNA from the Ovation protocol was labelled with the Encore Biotin Module (NuGen). Thirty-four microlitres (782 ng) of fragmented and labelled DNA (in 116 μl of Hybridization cocktail) was loaded on the Mouse Gene 2.0 ST gene chip (Affymetrix, Santa Clara, CA) and the chip was hybridized for 16–18 h in a 45 °C Affymetrix Gene Chip Hybridization Oven 645. The chip was stained and washed on an Affymetrix GeneChip Fluidics Station 450 using wash protocol FS450_0002. The chip was scanned on an Affymetrix Genechip Scanner 7G Plus at Molecular Biology Core Facilities at Dana-Farber Cancer Institute (DFCI). All arrays were normalized and processed in GenePattern (http://www.broadinstitute.org/cancer/software/genepattern/). Gene Set Enrichment Analysis (GSEA) was performed by using the Molecular Signatures Database v5.2 (http://software.broadinstitute.org/gsea/msigdb) and the standalone GSEA program (http://software.broadinstitute.org/gsea/downloads.jsp). Gene sets for mouse or human intrinsic subtypes were extracted from a recent study[37] by using upregulated genes (SAM fold change >1)

for each subtype, with SAM $q$-value of 0. Multiple Experiment Viewer (MeV) program (http://www.tm4.org/) was used to visualize the expression data and GSEA data.

**Mouse SNP array and analysis.** Total genomic DNAs from $Trp53$-null mammary tumours and from mammary glands (MGs) of a genetic background-matched (FVB) WT female were prepared by the AllPrep DNA/RNA Mini Kit (Qiagen). Mouse Diversity SNP array (Affymetrix) was used to determine genomic abnormalities in these tumours at Molecular Biology Core Facilities at DFCI. Normalization and data extraction from the scanned CEL files (together with >100 sample CEL files from inbred and F1 cross mice available from the Affymetrix website (http://www.affymetrix.com/catalog/prod100002/AFFY/Affymetrix%C2%AE-Mouse-Diversity-Genotyping-Array#1_3)) were performed by using the Affymetrix Genotyping Console Software 4.2 with the BRLMM-P algorithm. Strength of each SNP probe (average of log2 signals from both alleles) from each tumour sample was compared with that from the WT sample and the difference between these two was used to determine copy number alteration at each SNP locus.

**Quantitative RT–PCR and PCR.** Total RNAs from tumours were purified by either Trizol or the Allprep DNA/RNA mini kit (Qiagen). cDNA was generated with iScript (Bio-Rad, Berkeley, CA) according to the manufacturer's protocol. Quantitative RT–PCR (qRT–PCR, for RNA) and PCR (for genomic DNA) were performed using FastStart SYBR Green Master (Roche, Indianapolis, IN). PCR primers are listed below:

Primers for RT–PCR:
$Yap1$: forward: 5′-AATGCCGTCATGAACCCCAA-3′; reverse: 5′-GAGTGTCCCAGGAGAAACGG-3′
$Met$: forward: 5′-GCTCTGGAGGACAAGACCAC-3′; reverse: 5′-CGTGAAGTTGGGGAGCTGAT-3′
$Akt3$: forward: 5′-ACCGCACACGTTTCTATGGT-3′; reverse: 5′-CGGCTCGGCCATAGTCATTA-3′
$Gapdh$: forward: 5′-GGTGAAGGTCGGTGTGAACG-3′; reverse: 5′-CTCGCTCCTGGAAGATGGTG-3′

Primers for genomic DNA PCR:
$Yap1$: forward: 5′-AATGCCGTCATGAACCCCAA-3′; reverse: 5′-GCCAACTCCGCGAAAACAA-3′
$Met$: forward: 5′-CTCTGGAGGACAAGACCAC-3′; reverse: 5′-AAGGGCTGAGTCAGATCCCA-3′
$Akt3$: forward: 5′-ACCGCACACGTTTCTATGGT-3′; reverse: 5′-TGTCCCTACAGTCTCTGCAAAA-3′
$Lsd1$ (as a loading control for genomic DNA): forward: 5′-TTGAGTTGGTTGTGAGTCAC-3′; reverse: 5′-AGCGCTAACTTTAGAGCTGG-3′

**Statistics.** Student's $t$-test (two-tailed) was used to calculate the $P$ values. Data were reported as mean ± s.e.m. No statistical method was used to pre-determine the sample size for mice. No randomization or blinding was used in the in vivo studies.

**Data availability.** The authors declare that all data supporting the findings of this study are available within the article and its Supplementary Information files or from the corresponding author upon reasonable request. The microarray expression profiling data and SNP array data have been deposited in the Genbank GEO database (http://www.ncbi.nlm.nih.gov/geo/) under accession codes GSE77496 and GSE77497, respectively (also under the SuperSeries record number GSE77498).

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

## Acknowledgements

We thank Eunsil Park and Chia-Cheng Li for technical assistance. This research was supported by a Pathway-to-Independence K99/R00 grant from NCI (CA126980), a Seed Grant (SG-0062-10) and a Cancer Program Pilot Grant (DP-0137-13-00) from Harvard Stem Cell Institute, Milton Fund Award from Harvard University, Hearst Foundation Young Investigator Award and Start-up Fund from Brigham and Women's Hospital, and a Breakthrough Award from DOD (W81XWH-15-1-0100) to Z.L.

## Author contributions

L.T. designed and conducted experiments and co-wrote the manuscript. D.X. conducted experiments and analysed the data. Y.X. conducted experiments. R.T.B. performed histology evaluation. Z.L. designed experiments, analysed the data and wrote the manuscript.

## Additional information

**Competing financial interests**: The authors declare no competing financial interests.

**How to cite this article**: Tao, L. *et al.* Induced p53 loss in mouse luminal cells causes clonal expansion and development of mammary tumours. *Nat. Commun.* **8,** 14431 doi: 10.1038/ncomms14431 (2017).

**Publisher's note**: 

