## [Peer Review File · Nature Communications]

Reviewers' comments:

Reviewer #1 (Remarks to the Author): Expert in Breast cancer and stem cells

In this manuscript the authors use the Keratin 8 promoter to drive expression of Cre and thereby mediate recombination of floxed TP53 and YFP reporter alleles. The authors induced deletion of TP53 and activation of YFP via intraductal Ad-K8-Cre or using a germline K8-CreER transgene.

With regard to the adenoviral injection, it is unclear if the initially observed minor population of YFP positive cells actually expands. The authors first demonstration of labeling is at a time point 3-4 weeks after adenoviral injection. Thus, the reader has no idea how many cells were initially labeled (day 4-5 after injection for example). It would have been better if the authors first demonstrated by in situ (single molecule FISH for example) which cells had internalized the virus shortly after infection. This would be strengthened by Cre, p53 and YFP IHC at very short time points after injection.

The authors need to include more time points. If the p53 null terminally differentiated luminal cells do indeed expand, then one would see a chain of YFP+ cells next to each other that gets larger as time passes. The authors show a few labeled cells at 3-4 weeks, and then at 5 months, essentially all of the cells being YFP+ without any data in between. This does not illustrate a progressive process of terminally differentiated luminal epithelial expansion to the reader. Given that the K8 transgene may be expressed in luminal progenitors, the authors should clearly demonstrate that their construct is initially entirely limited to the mature terminally differentiated mammary epithelial cells (FACS sorting followed by analyses). K8 expression in mammary progenitors could explain most of the observations represented in this manuscript; the virus may integrate in those cells and the transgenic mouse model may also express CreER in the progenitors. The authors need to demonstrate beyond the shadow of a doubt that the expression is entirely limited to the mature terminally differentiated luminal mammary epithelium rather than luminal mammary progenitors in order for their argument to be valid.

If a process of p53 null luminal epithelial cell expansion does occur, where do the WT cells go? How does the proliferation of WT cells and TP53 deleted cells compare in vivo (BrdU staining)? What about cell death of non-deleted cells in vivo? Again, the authors would need more time points, controls and supporting data to clearly demonstrate expansion.

Beyond the technical deficiencies, if the p53 null cells do expand as suggested, this manuscript is largely a description of gene expression changes without demonstrating the importance of the genes from the lists that were generated. Also, the SNP data does not use an adequate number of independent biological replicates to power the conclusions associated with the general specificity of amplified loci when p53 is ablated. Which of the genes in the identified loci are functionally relevant for the observed tumor initiation?

With regard to the tumor data, it is not surprising that the loss of p53 eventually results in tumor formation. As cells proliferate they acquire mutations and without p53 to initiate DNA damage checkpoints mutant cells are allowed to propagate.

Given the amount of work necessary to clarify this manuscript, and the predominant reliance on gene expression or SNP data without detailed mechanistic insights, it is unclear how this manuscript can be revised in a way that would make it suitable for publication in this journal within a reasonable time frame.

Reviewer #2 (Remarks to the Author): Expert in Breast cancer and p53

In this study, the authors use two p53-null breast cancer models to understand the role of p53 loss in breast cancer development. First, they find that induced loss of p53 specifically in luminal cells leads to their clonal expansion via increased cell cycle activity and decreased apoptosis. Importantly, these changes do not alter their identity. The tumors that develop in 6-7 months after p53 loss were subject to microarray and SNP profiling. This allowed characterization of tumors into three classifications. The most important observations are 1) that Claudin-low tumors can originate from luminal cells and 2) the stem-like nature of p53-null proliferating cells. Specific comments:

1. On page 5, the authors indicate that 'luminal MECs apparently also led to aberrant alveolar cell expansion in virgin mice'. Should this be labeled data not shown?
2. Also on page 5, the authors indicate 'loss of p53 in luminal MECs leads to increased self-renewal of both ductal and alveolar luminal stem or progenitor cells'. How was self-renewal examined?
3. Parts of Figure 1a are somewhat misleading. The only p53 floxed allele that I know of deletes most p53 exons not just one as shown.
4. Please indicate how many ducts were counted in figure 1D.
5. In figures 3D, 4A and 4B, only 1 type 2 tumor and two type 1 tumors were arrayed. This is a very small number to be able to draw any conclusions.
6. The mice were purchased from Jackson Laboratory and the entire name of the allele should be provided in materials and methods so that readers know exactly which mice were purchased. The mice were also crossed to an FVB background and we need to know the number of times the mice were backcrossed and if marker analysis was performed to determine % of FVB.

Reviewer #3 (Remarks to the Author): Expert in cancer genomics

This manuscript reports the effects of p53 loss in luminal breast cells using lineage tracing and ancillary methods. It is of interest that p53 loss does not cause loss of differentiation, although some form of expansion of the p53-null population does occur. It is a pity that the authors do not address the issue of whether this is a clonal expansion, and whether loss of p53 is sufficient as a driver versus p53 loss being permissive for secondary driver mutations. To their credit, the authors do perform some limited analyses of this type for the tumours that develop in their p53 mutants, and the recurrent copy number changes they find are noteworthy. In an ideal world, mice carrying those copy number changes alone, without p53 mutation, could be studied, or crosses with knockouts for genes in the regions could be performed - but I guess this is for follow-on work. The chr6 CNA is not very convincing, as presented. It would be good for this (and 9q) to see each tumour separately in Fig 4.

Responses to Reviewers' comments:

Reviewers' comments:

Reviewer #1 (Remarks to the Author):Expert in Breast cancer and stem cells

In this manuscript the authors use the Keratin 8 promoter to drive expression of Cre and thereby mediate recombination of floxed TP53 and YFP reporter alleles. The authors induced deletion of TP53 and activation of YFP via intraductal Ad-K8-Cre or using a germline K8-CreER transgene.

With regard to the adenoviral injection, it is unclear if the initially observed minor population of YFP positive cells actually expands. The authors first demonstration of labeling is at a time point 3-4 weeks after adenoviral injection. Thus, the reader has no idea how many cells were initially labeled (day 4-5 after injection for example). It would have been better if the authors first demonstrated by in situ (single molecule FISH for example) which cells had internalized the virus shortly after infection. This would be strengthened by Cre, p53 and YFP IHC at very short time points after injection.

Response: We thank the Reviewer for pointing this out. To address this concern, we characterized mammary glands from these conditional mice three days after adenoviral injection by both co-immunofluorescence (co-IF) staining and FACS analysis. By co-IF staining of YFP and Cre recombinase (expressed from adenovirus), we found that Cre-expressing cells largely overlapped with YFP-marked cells and these cells were localized in mammary ducts (new Supplementary Fig. 1a). To determine the subtypes of mammary luminal cells targeted by *Ad-K8-Cre* virus, we performed FACS analysis based on a strategy developed by Stingl and colleagues that separated luminal cells as three luminal subpopulations based on Sca1 and CD49b staining (*Shehata et al., 2012, PMID: 23088371*); we found that three days after *Ad-K8-Cre* injection, all these three luminal subpopulations [i.e., Sca1-CD49b+ luminal progenitors (LPs) (ER-), Sca1+CD49b+ LPs (ER+), and Sca1+CD49b- non-clonogenic luminal (NCL) cells (largely ER+)] could be genetically marked by *Ad-K8-Cre* (new Supplementary Fig. 1b-c).

To provide evidence that the initially observed YFP+ luminal cells (i.e., 3 days post-injection) indeed expanded upon p53-loss, we counted YFP-marked clones (as single cell clones or clones with at least 2 YFP+ cells) in mammary gland sections from mice 3 days (initial labeling) or 3-4 weeks (short-term chase) after injection. From this analysis, we found that in wild-type (WT) mice, the percentages of single cell clones and multi-cell clones did not change significantly when compared mammary glands after the short-term chase to those at the initial labeling stage; however, in induced p53-mutant mice, there was a significant increase in the percentages of multi-cell clones after chasing for 3-4 weeks (compared to those at the initial labeling stage and to WT controls). Representative co-IF staining pictures showing similar YFP-labeling patterns in WT and p53 mice at the initial labeling stage are shown in the revised Fig. 1b (left panels), together with those showing multi-cell clones in p53 mice after the short-term chase (Fig. 1b, middle panels). Quantification data for single-cell and multi-cell clones at these two time points are shown in the new Fig. 1c. In addition to the clonal analysis, our FACS analysis for the YFP-marked population also revealed that 3-4 weeks after injection, the population sizes of YFP+ cells from the induced p53-mutant mice were often larger than those from the matched WT mice from the same experiment (quantification data shown in the new Fig. 1f). All together, these data supported that upon induced p53-loss, the initially YFP-labeled luminal cells (at least a portion of them) indeed expanded.

The authors need to include more time points. If the p53 null terminally differentiated luminal cells do indeed expand, then one would see a chain of YFP+ cells next to each other that gets larger as time passes. The authors show a few labeled cells at 3-4 weeks, and then at 5 months, essentially all of the cells being YFP+ without any data in between. This does not illustrate a progressive process of terminally differentiated luminal epithelial expansion to the reader. Given that the K8 transgene may be expressed in luminal progenitors, the authors should clearly demonstrate that their construct is initially entirely limited to the mature terminally differentiated mammary epithelial cells (FACS sorting followed by analyses). K8 expression in mammary progenitors could explain most of the observations represented in this manuscript; the virus may integrate in those cells and the transgenic mouse model may also express CreER in the progenitors. The authors need to demonstrate beyond the shadow of a doubt that the expression is entirely limited to the mature terminally differentiated luminal mammary epithelium rather than luminal mammary progenitors in order for their argument to be valid.

Response: We agree with the Reviewer that by including more time points, they may make our data like more convincing. However, as the Reviewer also mentioned below, due to loss of p53, cells may acquire mutations and some of the acquired secondary mutations may drive clonal expansion of *Trp53*-null cells. We reasoned that shortly after induced loss of p53, the phenotype is most likely caused by p53-loss alone, but at later time points, we cannot rule out any potential contribution from acquired secondary mutations to the phenotype. Therefore, our strategy was to “catch” the earliest time point after induced p53-loss at which we could already observe a phenotype. We found 3-4 weeks after induction was such a time point and thus focused our efforts on this time point. Although we could include more time points between 3-4 weeks and 5 months after induction, we worried that at these “intermediate” time points, we could not rule out the possibility that some acquired secondary mutations (due to p53-loss) might drive clonal expansion of *Trp53*-null cells. In fact, although we showed extensive expansion of YFP-marked luminal cells in mammary glands from mutant mice >5 months after induced p53 loss (Fig. 1b, right panels), we mainly want to show the long-term consequence of p53-loss in mammary luminal cells and have toned down our previous implication that this long-term phenotype was due to p53-loss alone, as we cannot rule out a contribution of acquired secondary mutation(s) to it at this late stage.

As to the Reviewer’s comment: “K8 expression in mammary progenitors could explain most of the observations represented in this manuscript”, we showed previously (*Tao et al., 2014, PMID: 24936465*) that since K8 is a pan-luminal marker, K8 promoter-driven Cre expression could target both luminal progenitors (LPs) and mature luminal cells. In addition, the Blanpain group, who made the *K8-CreER* mouse line, also showed that K8 could target both LPs (i.e., unipotent luminal stem cells) and mature luminal cells (*van Keymeulen et al., 2011, PMID: 21983963*). As described above, to further determine the type of mammary luminal cells targeted by our *Ad-K8-Cre* virus, we performed FACS analysis of the YFP-marked population shortly after *Ad-K8-Cre* injection and confirmed that K8 could target all three luminal subpopulations, including the ER+ and ER- LP subpopulations and the NCL subpopulation, which is enriched with ER+ mature luminal cells (MLs) (based on *Shehata et al., 2012, PMID: 23088371*). Thus, clearly these K8-based lineage-tracing tools target both mammary progenitors and MLs. If as the Reviewer stated, the luminal expansion phenotype we described in this manuscript was due to K8 expression in mammary progenitors (i.e., these progenitors produce daughter cells and as a result, form large clones), we would expect to

observe a similar phenotype from WT control mice as well. However this is not the case. In each experiment, we always included matched WT controls (e.g., mice with *R26Y* reporter alone). Upon the long-term chase (>5 months), although we also observed large clones (possibly clones formed by mammary progenitors) in WT control mice, the frequency of such clones was much lower compared to that of p53-mutant mice (Fig. 1d).

Please note that in lineage-tracing experiments based on either *Ad-K8-Cre* injection or *K8-CreER* (upon tamoxifen induction), due to the transient nature of genetic marking, typically only a portion of the K8+ luminal cells are labeled, thus producing a mosaic system. In WT mice, YFP-marked and non-marked luminal cells are largely equipotent cells and due to constraints of the mammary ductal architecture, YFP+ and YFP- luminal cells are expected to undergo neutral clonal competition in a stochastic fashion (i.e., both YFP+ and YFP- mammary progenitors may develop into a large clone with an equal chance). Since in our experiments, we only targeted a small number of luminal cells (e.g., after “pulse” labeling by *Ad-K8-Cre*, only ~1% luminal cells were YFP+, the remaining 99% cells were YFP-), the chance for YFP+ progenitors to develop into large clones that fill the entire duct should be very low, as demonstrated from the clonal phenotype of long-term chase in our WT control mice (Fig. 1b, right panels, top pictures, and quantification in Fig. 1d), as well as from that in WT *K8-CreER* mice, reported previously by the Blanpain group (*van Keymeulen et al., 2011, PMID: 21983963*, in this case, a small portion of luminal cells (<10%) were labeled initially too). In contrast, in p53-mutant mice, YFP+ luminal cells are *Trp53*-null, and YFP- luminal cells are *Trp53*-WT; thus here YFP+ and YFP- cells are not equipotent cells any more and loss of p53 in YFP+ cells appears to shift the balance of neutral clonal competition in favor of YFP+ *Trp53*-null cells, leading to their clonal expansion; in another word, the chance for them (whether they are progenitors or even mature luminal cells) to undergo clonal expansion and develop into large clones is much higher. Thus, although we only targeted ~1% or less luminal cells at the initial stage, due to p53-loss (and possibly also due to acquired secondary mutations at later stages), these *Trp53*-null luminal cells had a higher chance to outcompete their YFP- WT neighbor cells and eventually developed into large clones.

Lastly, we wish to also point out that the current definition of mammary progenitors is largely based on *in vitro* clonogenic arrays. For instance, ER- LPs (Sca1-CD49b+) and ER+ LPs (Sca1+CD49b+) are two types of LPs that can form colonies *in vitro* (*Shehata et al., 2012, PMID: 23088371*), but whether they indeed give rise to terminally differentiated luminal cells *in vivo* under the physiological setting

has not been formally demonstrated. We and others have shown that ER- LPs mainly represent alveolar progenitors and undergo alveolar differentiation and expansion during pregnancy and lactation (Tao et al., 2015, PMID: 26120057; Chang et al., 2014, PMID: 24398145). Recently, by analyzing cell division kinetics, the Stingl group provided evidence to support that the NCL subpopulation, which includes the majority of ER+ luminal cells, may be self-sustained by its restricted progenitors (Girardi et al., 2015, PMID: 26511661). The exact identity of these NCL progenitors is unknown but they appear to be ER- luminal cells. Interestingly, when characterizing the luminal phenotype at the early time point after induced p53-loss, we found evidence of increased expansion of ER+ luminal cells; the YFP-marked multi-cell clones observed in p53-mutant mice 3-4 weeks after induction often contained multiple ER+ luminal cells, but also ER- luminal cells (new Fig. 2f), suggesting that at this early time point, one of the main effects of p53-loss on the luminal lineage may be to increase clonal expansion of the NCL subpopulation. Thus, although we cannot show that K8-based genetic marking is only limited to the mature terminally differentiated luminal mammary epithelium (MLs), we can show that it could target the NCL subpopulation, which includes ER+ MLs (new Supplementary Fig. 1b-c), and induced p53-loss in the K8+ luminal lineage could lead to expansion of these ER+ cells (new Fig. 2e-f).

If a process of p53 null luminal epithelial cell expansion does occur, where do the WT cells go? How does the proliferation of WT cells and TP53 deleted cells compare in vivo (BrdU staining)? What about cell death of non-deleted cells in vivo? Again, the authors would need more time points, controls and supporting data to clearly demonstrate expansion.

Response: We measured proliferation and apoptosis of luminal cells at the above-described early time point after induce p53-loss. We quantified their proliferation by Ki67 staining and found that at this time point, YFP+ *Trp53*-null luminal cells contained more Ki67+ cells than stage-matched YFP+ WT luminal cells (new Fig. 2c and Supplementary Fig. 2b). We performed apoptosis study by TUNEL staining; however, we found that in mammary ducts of virgin females, the number of apoptotic cells was very low; therefore we were unable to score enough YFP+TUNEL+ cells to make a meaningful conclusion. Thus, although we cannot provide a definitive explanation for what happened with *Trp53*-null cells vs. their WT neighbor cells, our data suggest that expansion of *Trp53*-null luminal cells is in

part due to their increased proliferation and cell cycle activity; this conclusion is also supported by our microarray data (Fig. 2a).

Beyond the technical deficiencies, if the p53 null cells do expand as suggested, this manuscript is largely a description of gene expression changes without demonstrating the importance of the genes from the lists that were generated. Also, the SNP data does not use an adequate number of independent biological replicates to power the conclusions associated with the general specificity of amplified loci when p53 is ablated. Which of the genes in the identified loci are functionally relevant for the observed tumor initiation?

Response: We agree with the Reviewer that it would greatly strengthen this manuscript if we could further demonstrate the functional importance of some of the differentially expressed genes in luminal cells upon p53-loss. However, this task is certainly not trivial and may require additional years of time to finish. There are similar studies (in scope) published recently in *Nature* (*van Keymeulen et al., 2015, PMID: 26266985; Koren et al., 2015, PMID: 26266975*) describing acquisition of a multipotent mammary stem cell-like phenotype by luminal cells upon ectopic expression of PIK3CA^{H1047R}. In these studies, the authors also described gene expression changes in response to PIK3CA^{H1047R} expression without further demonstrating the importance of these genes for the phenotype. In addition, these studies demonstrated that induced loss of p53 could accelerate the PIK3CA^{H1047R}-induced cancer phenotype. However, it was unclear what was the role of p53 in this process and whether p53-loss could also contribute to the acquired multipotency phenotype. As *TP53* and *PIK3CA* are the two most commonly mutated genes in human breast cancer, here we mainly wanted to complement these two studies by providing a timely communication to report the phenotype of induced loss of p53 alone under a similar setting (i.e., in K8+ luminal cells). In addition to providing a description of gene expression changes upon p53-loss, we have also validated some of the key changes, such as signatures showing increased cell cycle activity (validated in Fig. 2c and Supplementary Fig. 2b) and enrichment of the mature luminal cell/estrogen receptor-related program (validated in Fig. 2f). The increased proliferation of *Trp53*-null luminal cells may be related to downregulation of the p53 target gene *Cdkn1a* (p21) (Fig. 2d); but this is probably as expected.

As to the SNP data, although we have only profiled 5 tumors, we have validated both genomic amplification and overexpression of three key genes identified from the SNP data, *Yap1*, *Met* and *Akt3*, in many more tumors from both the *Ad-K8-Cre* and *K8-CreER* cohorts (Fig. 4c-d and Supplementary Fig. 4d). These validation experiments confirmed our key findings from the SNP data. Although we did not directly test whether ectopic expression of these three genes in luminal cells, either alone or in combination with p53-loss, would lead to initiation of their corresponding mammary tumor subtypes, there is evidence in literature that has linked their ectopic expression to their corresponding tumor phenotypes (e.g., *Yap1* to EMT, *Met* and *Akt3* to Basal-Like breast cancer).

With regard to the tumor data, it is not surprising that the loss of p53 eventually results in tumor formation. As cells proliferate they acquire mutations and without p53 to initiate DNA damage checkpoints mutant cells are allowed to propagate.

Response: Although it is not surprising that loss of p53 eventually leads to mammary tumor formation, largely due to secondary mutations acquired by *Trp53*-null cells, our tumor data has provided several novel points: 1) Although loss of p53 does not alter the fate of mammary luminal cells directly (which by itself is a novel finding), it opens up the “gate” for luminal cells to lose their original luminal identity, leading to development of mammary tumors with loss of luminal epithelial features. 2) Although it has been proposed that Claudin-Low breast cancer originates from transformation of basal mammary cells (i.e., basal mammary stem cells), our data has provided direct evidence to support that this breast cancer subtype can also have a luminal origin.

Given the amount of work necessary to clarify this manuscript, and the predominant reliance on gene expression or SNP data without detailed mechanistic insights, it is unclear how this manuscript can be revised in a way that would make it suitable for publication in this journal within a reasonable time frame.

Response: As explained above, although we could not provide additional mechanistic insights for how induced p53-loss in luminal cells contributes to breast tumorigenesis within the 3-month revision time frame, we wish to point out again that our main purpose for publishing this study is to provide a timely report of how induced loss of p53 in mammary luminal cells affects this lineage, leading to

development of breast cancer. We believe our study has led to several novel findings: 1) Induced loss of p53 in the luminal lineage leads to increased proliferation and clonal expansion of luminal cells, in particular ER+ ductal luminal cells. 2) Induced loss of p53 alone in luminal cells does not alter their luminal cell fate directly, but facilitates acquisition of stem cell-like properties in them, in a way similar to the role of p53 in (blocking) cellular reprogramming. 3) Claudin-Low breast cancer can have a luminal origin. These novel findings clarify roles of p53 in the acquisition of “stemness” by breast cancer cells and enhance our understanding of cells of origin of breast cancer.

Reviewer #2 (Remarks to the Author): Expert in Breast cancer and p53

In this study, the authors use two p53-null breast cancer models to understand the role of p53 loss in breast cancer development. First, they find that induced loss of p53 specifically in luminal cells leads to their clonal expansion via increased cell cycle activity and decreased apoptosis. Importantly, these changes do not alter their identity. The tumors that develop in 6-7 months after p53 loss were subject to microarray and SNP profiling. This allowed characterization of tumors into three classifications. The most important observations are 1) that Claudin-low tumors can originate from luminal cells and 2) the stem-like nature of p53-null proliferating cells.

Specific comments:

1. On page 5, the authors indicate that 'luminal MECs apparently also led to aberrant alveolar cell expansion in virgin mice'. Should this be labeled data not shown?

Response: This actually refers to the expansion of YFP+ alveolar cells in the induced p53-mutant mice after a long-term chase (yellow arrows in the current Fig. 1b, right panels). In the revised manuscript, we have changed the way we present these lineage-tracing data and this phenotype is now described in the first paragraph in page 6, with proper label of the figure panel.

2. Also on page 5, the authors indicate 'loss of p53 in luminal MECs leads to increased self-renewal of both ductal and alveolar luminal stem or progenitor cells'. How was self-renewal examined?

Response: As our *in vivo* study does not directly test self-renewal of stem and progenitor cells, we have removed this sentence.

3. Parts of Figure 1a are somewhat misleading. The only p53 floxed allele that I know of deletes most p53 exons not just one as shown.

Response: We thank the Reviewer for pointing this out. We have revised this figure panel accordingly to reflect deletion of multiple exons from the floxed *Trp53* conditional knockout allele.

4. Please indicate how many ducts were counted in figure 1D.

Response: We have provided this information in the corresponding figure legend (page 28, under **(d)**, “In both **(c)** and **(d)**, at least 10 ducts per section were counted”).

5. In figures 3D, 4A and 4B, only 1 type 2 tumor and two type 1 tumors were arrayed. This is a very small number to be able to draw any conclusions.

Response: We have included more samples in our microarray study, including one more sample of *Trp53*-null luminal cells and three more tumor samples (E2, E7, E9, thus including all tumor samples that had been profiled by the SNP array). These additional samples have further strengthened our original conclusions and all the corresponding figures have been updated to reflect the new microarray data.

6. The mice were purchased from Jackson Laboratory and the entire name of the allele should be provided in materials and methods so that readers know exactly which mice were purchased. The mice were also crossed to an FVB background and we need to know the number of times the mice were backcrossed and if marker analysis was performed to determine % of FVB.

Response: We have provided the entire names of all the mouse alleles in the Materials and Methods section. We have also provided the FVB backcrossing information there (“All mice were backcrossed to the FVB/N background for at least 6 generations”, page 16). Based on our SNP array data, the rates for homozygous SNP markers for our profiled tumors E2, E4, E5, E8, E9, and WT control (pure FVB)

are 97.96%, 97.61%, 96.98%, 97.23%, 97.88%, and 98.48%, respectively. Thus, they were derived from our backcrossed mice that were very close to a pure FVB background.

Reviewer #3 (Remarks to the Author):Expert in cancer genomics

This manuscript reports the effects of p53 loss in luminal breast cells using lineage tracing and ancillary methods. It is of interest that p53 loss does not cause loss of differentiation, although some form of expansion of the p53-null population does occur. It is a pity that the authors do not address the issue of whether this is a clonal expansion, and whether loss of p53 is sufficient as a driver versus p53 loss being permissive for secondary driver mutations. To their credit, the authors do perform some limited analyses of this type for the tumours that develop in their p53 mutants, and the recurrent copy number changes they find are noteworthy. In an ideal world, mice carrying those copy number changes alone, without p53 mutation, could be studied, or crosses with knockouts for genes in the regions could be performed - but I guess this is for follow-on work. The chr6 CNA is not very convincing, as presented. It would be good for this (and 9q) to see each tumour separately in Fig 4.

Response: To address the issue of clonal expansion, we have performed several additional experiments, including characterization of YFP-marked mammary epithelial cells shortly after *Ad-K8-Cre* adenoviral injection (3-days post-injection as the initial labeling stage, shown in revised Fig. 1b, left panels), and further characterization of YFP-marked cells in mammary glands 3-4 weeks post-injection (i.e., short-term chase) [e.g., Ki67 staining (new Fig. 2c and Supplementary Fig. 2b), ER staining (new Fig. 2f)]. By comparing YFP-marked clones at the initial labeling stage to those after the short-term chase, we found that the portion of multi-cell clones was significantly increased in the induced p53-mutant mammary glands upon the short-term chase (new Fig. 1c); we also found that at this stage, there were more YFP+ Ki67+ cells in the p53-mutant mammary glands (new Fig. 2c and Supplementary Fig. 2b) and the *Trp53*-null YFP+ multi-cell clones exhibited expansion of ER+ ductal luminal cells (new Fig. 2f). All these new data provide support for clonal expansion of *Trp53*-null luminal cells. Since this phenotype was observed in mice as early as ~3 weeks after induced p53-loss, there was probably insufficient time for these *Trp53*-null luminal cells to acquire secondary mutations (due to p53-loss) that could consistently drive their expansion. Thus, we believe loss of p53 is sufficient as a driver for clonal expansion of luminal cells, but certainly not sufficient as a driver for

the development of full-blown mammary tumors; instead, recurrent secondary mutations acquired by luminal cells (due to p53-loss), such as amplification of *Yap1* or *Met*, are more likely to be drivers for those mammary tumors observed in this mouse model. Lastly, based on the Reviewer's suggestion, we have changed the style of presenting the recurrent genomic lesions as individual plots for each tumors (in the revised Fig. 4a and Supplementary Fig. 4b-c). We hope this change would make it easier for reviewers and readers to have a quick overview of our SNP array data.

Reviewers' comments:

Reviewer #2 (Remarks to the Author):

The authors responded well to my comments for the most part. I was disappointed that they did not directly test self-renewal of stem and progenitor cells and have backed off on that conclusion. Still the studies are interesting.

Reviewer #3 (Remarks to the Author):

I am generally happy with the authors' response to my comments, although I do not agree that additional mutations have to occur within the window between Cre activation and sampling. Some may be pre-existing. This is a critical aspect of the manuscript, and should at least be discussed.

Reviewer #4 (Remarks to the Author):

This is an interesting study but it needs to be further extended before concluding that claudin-low tumors have a luminal origin. Several comments have been addressed but there are remaining issues on tumor biology and interpretation of data.

1. The study uses a single gene promoter (K8) to target luminal cells. To make the claim that luminal cells are the cellular origin of claudin-low tumors, it is essential to use a second luminal-specific promoter for p53 deletion to confirm the observations. A basal-specific promoter also would be very helpful for data interpretation, as recently performed in studies by Koren et al Nature and van Keymeulen et al Nature 2015, where they reported mesenchymal, claudin-low type tumors with two different basal promoters.
2. Although increased, too few tumors have been analyzed for RNA expression. Only three Type 1 and one Type 2 tumors were analyzed (and 2 mixed, but this is difficult to interpret). Why are the Type 2 tumors so small when basal-like cancers are very aggressive and grow rapidly?
3. p53 heterozygous and null mouse models give a spectrum of tumors, of which claudin-low is a minority (eg Herschkowitz et al PNAS 2012). The same occurs in the case of human breast cancer. This study predominantly sees carcinosarcomas that are rare in humans – these were shown to have molecular properties of claudin-low tumors, but may relate to the specific mouse model. Why were luminal tumors not observed given that the luminal lineage promoter K8 was used?
4. The conclusion that claudin-low tumors have a luminal origin is premature. The data show that p53 loss (in luminal cells) facilitates the acquisition of MaSC-like properties. This study does not exclude the possibility that luminal cells assume basal-like features later in neoplastic progression and that it is these basal-like cells from which tumors develop. Samples were isolated at an early time (4 weeks after Tam injection) but this acquisition may occur subsequently during progression. ie there is no direct evidence for the major conclusion.

Responses to Reviewers' comments (Manuscript # NCOMMS-16-04284A):

Reviewer #2 (Remarks to the Author):

The authors responded well to my comments for the most part. I was disappointed that they did not directly test self-renewal of stem and progenitor cells and have backed off on that conclusion. Still the studies are interesting.

Response: We will certainly continue this work and will test whether induced loss of p53 in luminal cells alters the self-renewal properties of luminal stem and progenitor cells [either alone or together with other oncogenic event(s)] in future studies.

Reviewer #3 (Remarks to the Author):

I am generally happy with the authors' response to my comments, although I do not agree that additional mutations have to occur within the window between Cre activation and sampling. Some may be pre-existing. This is a critical aspect of the manuscript, and should at least be discussed.

Response: We agree with this Reviewer that we cannot rule out the possibility of pre-existing mutations in the mice we used for this study (FVB background) even before Cre activation. As mammary tumors developed in these mice were more homogeneous (i.e., predominantly Claudin-Low tumors) and had a shorter latency (~6-9 months) than those from other mouse models with p53-loss (e.g., in the BALB/c transplantation model, the average latency of tumor development was ~12 months), it is certainly possible that pre-existing mutations in our mice might have also contributed to quick development of the relatively homogeneous tumors in our model. We have included this point in the Discussion (Paragraph 3).

Reviewer #4 (Remarks to the Author):

This is an interesting study but it needs to be further extended before concluding that claudin-low tumors have a luminal origin. Several comments have been addressed but there are remaining

issues on tumor biology and interpretation of data.

1. The study uses a single gene promoter (K8) to target luminal cells. To make the claim that luminal cells are the cellular origin of claudin-low tumors, it is essential to use a second luminal-specific promoter for p53 deletion to confirm the observations. A basal-specific promoter also would be very helpful for data interpretation, as recently performed in studies by Koren et al Nature and van Keymeulen et al Nature 2015, where they reported mesenchymal, claudin-low type tumors with two different basal promoters.

Response: We agree with this Reviewer that it may make our paper stronger by including a second luminal-specific promoter as well as a basal-specific promoter. However, to our knowledge, there are no other well-characterized luminal-specific promoters that are currently available and can target the entire luminal lineage as specific as the *K8* promoter. *K18* promoter was used previously to target luminal cells but it appears to target largely mature luminal cells (Van Keymeulen et al, 2011, PMID: 21983963). Other commonly used promoters for mammary epithelial cells (MECs) include the *MMTV* promoter, which targets both luminal and basal cells, and the *Wap* or *Blg* promoter, which targets ER⁺ luminal progenitors/alveolar cells. As to the basal-specific promoter (e.g., *K5* or *K14*), the main concern we have is that whether basal cells make any significant contribution to the luminal lineage in adult mammary glands *in vivo* remains an open question: whereas some *in vivo* studies suggested that basal cells do not contribute to the luminal lineage in adults (e.g., Van Keymeulen et al, 2011, PMID: 21983963; Wuidart et al, 2016, PMID: 27284162), others provided evidence to support significant contribution of basal cells to luminal cells *in vivo* (e.g., Rios et al, 2014, PMID: 24463516; Wang et al, 2015, PMID: 25327250). Based on our own study, we showed previously that intraductal injection of *Ad-K14-Cre* adenovirus led to genetic marking of both basal cells and a portion of luminal cells (Tao et al, 2014, PMID: 24936465). We recently tested the *Ad-K5-Cre* adenovirus and found that although it appears to be more specific for the basal lineage than *Ad-K14-Cre*, it also led to genetic marking of a small number of MECs in the luminal gate (from flow cytometry). Due to the limitation of genetic marking based on fluorescent protein reporters (spatial clonal interference, i.e., one cannot definitively determine whether cells that are spatially close to each other are derived from a single founder cell or from multiple founders that are just close to each other by chance), it remains unsolved as to whether basal MECs normally

contribute to the luminal lineage. Thus if we induce p53-loss in MECs using either *Ad-K14-Cre* or *Ad-K5-Cre* or using the *K5-CreER* transgenic model (upon tamoxifen induction), due to genetic marking of a small number of luminal MECs (either due to “leakiness” of the *K14* or *K5* promoter, or due to normal differentiation of marked basal cells to luminal cells), we cannot say for sure whether any resulting mammary tumors have a clear basal origin or both a basal and a luminal origin. For the purpose of this paper, we wanted to focus on addressing the question of how induced loss of p53 in luminal MECs, which are more committed MECs and may be the cell-of-origin for most breast cancers, affects the luminal lineage, leading to development of mammary tumors. Although we only used a single luminal-specific promoter, the *K8* promoter, we applied both the intraductal injection approach (based on the *Ad-K8-Cre* adenovirus) and the tamoxifen-induced *K8-CreER* transgenic approach. Since both approaches led to similar results and since the specificity of the *K8* promoter for the mammary luminal lineage has been quite well established in the literature (e.g., *Van Keymeulen et al, 2011* and *2015*, PMIDs: 21983963 and 26266985; *Koren et al, 2015*, PMID: 26266975; *Tao et al, 2014*, PMID: 24936465), we feel it is reasonable to claim that mammary tumors developed in these mice were originated from luminal cells. Interestingly, a previous study showed that induced expression of *Met* (i.e., also a recurrent genomic lesion identified in our tumor model) and disruption of p53 (by *MMTV-Cre*) led to development of predominantly Claudin-Low tumors (*Knight et al, 2013*, PMID: 23509284); although due to the use of the *MMTV* promoter, one cannot claim Claudin-Low tumors developed in this model had a clear luminal origin, at least this study is consistent with our observation. In studies by *Koren et al* and *van Keymeulen et al* (*Koren et al, 2015*, PMID: 26266975; *Van Keymeulen et al, 2015*, PMID: 26266985), although they reported mesenchymal, claudin-low type tumors with two different basal promoters, the models were mainly based on ectopic expression of the *PIK3CA*^{H1047R} oncogene, rather than based on p53-loss alone. Lastly, as explained in more details below, although we showed development of Claudin-Low tumors from *K8*⁺ luminal cells in our model, these tumors most likely progressed through a basal-like state first; thus our observation is actually consistent with the prevailing view of a basal origin for Claudin-Low breast cancer. Practically, any additional experiments using a different promoter would require another year of work (depending on tumor latency), which is unfortunately not feasible for the revision that we hope to complete within the typical 3-months time window.

2. Although increased, too few tumors have been analyzed for RNA expression. Only three Type

1 and one Type 2 tumors were analyzed (and 2 mixed, but this is difficult to interpret). Why are the Type 2 tumors so small when basal-like cancers are very aggressive and grow rapidly?

Response: First, please note we have revised our classification of these tumors: instead of having a category of Type1/2-mixed tumors (i.e., tumors with both K8⁻K14⁻ mesenchymal-like cells and K8⁺K14⁺ basal-like cells), we have grouped them into the Type 1 tumor category. The reason for this is that after further characterization, we found that the majority of tumors developed in our model all contained K14⁺ basal-like tumor cells (either K14⁺K8⁺ or K14⁺K8⁻ cells), and to a lesser degree, K5⁺ cells (new Supplementary Fig. 3e-f and Supplementary Table 1); thus we feel the previously described Type1/2-mixed tumors may just represent Type 1 tumors with different numbers of basal-like cells, rather than a separate subtype. Our main criterion to assign each tumor to Type 1 or Type 2 group is based on their morphologies: Type 1 tumors contain mainly mesenchymal-like, spindle-appearing tumor cells, whereas Type 2 tumors contain largely epithelial-like, round-appearing tumor cells. To address this Reviewer's concern here, we have profiled several more tumors (from the *K8-CreER*/tamoxifen approach; previously profiled tumors were all from the *Ad-K8-Cre* injection approach). Based on the new simple Type 1 and Type 2 categorization scheme, overall we have profiled eight Type 1 tumors and one Type 2 tumor. We then compared their similarities to human and mouse intrinsic subtypes individually. We found that almost all these tumors, even including the Type 2 tumor, exhibited the highest molecular similarities to the Claudin-Low subtype, followed by the Basal-Like subtype (new Fig. 3d-e). However, the Type 2 tumor indeed exhibited the lowest and highest enrichment scores for the Claudin-Low and Basal-Like subtypes, respectively (new Fig. 3d-e and Supplementary Table 2). Thus, it appears our inducible model mainly led to development of Claudin-Low tumors. As we showed that induced loss of p53 in luminal cells only led to their clonal expansion, we reasoned that most likely it was the acquired recurrent secondary mutations [e.g., *Yap1* and/or *Met* amplification(s)] that drove development of these Claudin-Low tumors. In the Discussion section, we also discussed other possibilities to explain why mammary tumors from our model were more homogeneous: for instance, this could also be due to the genetic background of our induced mice (i.e., the FVB background). Please note that the “smaller” Type 2 tumors observed in our mice were based on comparison to the “larger” Type 1 tumors. The majority of Type 1 tumors were the largest tumor we detected in each induced female and due to its large size and short latency, we had to euthanize the affected mouse and the Type 2 tumors

were typically detected in the remaining injected mammary glands. We suspect that if given enough time, these Type 2 tumors (or even other subtypes of tumors, such as luminal tumors) may develop further and become large tumors as well; alternatively, it is also possible that these Type 2 tumors may represent an intermediate basal-like state in their way to eventually develop into the Type 1 tumors, rather than a stable tumor subtype. As mammary tumors from our model appear to be quite homogeneous [e.g., almost all of them carried *Yap1* and/or *Met* amplification(s), based on our PCR analyses], even if we were to profile all of them, we do not expect to reveal any additional mammary tumor subtype(s) from them.

3. p53 heterozygous and null mouse models give a spectrum of tumors, of which claudin-low is a minority (eg Herschkowitz et al PNAS 2012). The same occurs in the case of human breast cancer. This study predominantly sees carcinosarcomas that are rare in humans – these were shown to have molecular properties of claudin-low tumors, but may relate to the specific mouse model. Why were luminal tumors not observed given that the luminal lineage promoter K8 was used?

Response: Please refer to our response to the Critique #2 above. As induced loss of p53 in luminal cells alone did not cause mammary tumor initiation directly, we reason that it is the acquired secondary mutations (in cooperation with p53-loss) that may not only drive development of mammary tumors, but also determine the tumor subtypes. Since almost all tumors we identified in our mice carried *Yap1* and/or *Met* amplification(s), which appear to lead to Claudin-Low tumors when under the p53-loss background, we suspect this could be the main reason to explain why we mainly detected Claudin-Low tumors, but not luminal tumors, in our model. Alternatively, the FVB background of our induced mice might also contribute to this. This could be due to unique genetic modifiers or pre-existing mutations in FVB mice. Our induced mice with the FVB background developed large Claudin-low tumors with a quite short latency (i.e., ~6-9 months), which required euthanization of the mice, so that there might be no sufficient time for other slower growing mammary tumors (e.g., luminal tumors) to fully develop; in contrast, heterogeneous mammary tumors developed in the BLBC/c transplantation model had a considerably longer latency [in average ~12 months: most tumors had a latency of around 40 weeks (~9 months) to 60 weeks (~14 months), based on *Herschkowitz et al, PNAS 2012*].

4. The conclusion that claudin-low tumors have a luminal origin is premature. The data show that p53 loss (in luminal cells) facilitates the acquisition of MaSC-like properties. This study does not exclude the possibility that luminal cells assume basal-like features later in neoplastic progression and that it is these basal-like cells from which tumors develop. Samples were isolated at an early time (4 weeks after Tam injection) but this acquisition may occur subsequently during progression. ie there is no direct evidence for the major conclusion.

Response: We previously concluded that Claudin-Low tumors in our model had a luminal origin. We did not intend to imply that Claudin-Low tumors developed from luminal MECs directly; in fact, we have already shown some data to suggest that although these Claudin-Low tumors were derived from luminal cells, they probably progressed through a basal-like intermediate state first, similar to what the Reviewer has suggested here. In the revised manuscript, we have further strengthened this point: we have further characterized all the tumors we collected by staining for basal markers K5 and K14, and found that the majority of them (including those Claudin-Low tumors) contained K14⁺ (and to a lesser degree, K5⁺) basal-like tumor cells at varying degrees (new Supplementary Fig. 3e-f and Supplementary Table 1). Importantly, we also found that almost all histologically aberrant premalignant lesions contained cells with luminal-to-basal change (i.e., upregulation of K14 or K5, Supplementary Fig. 3a-c). These data suggest that the majority of these tumors probably progressed through a basal-like intermediate state before they became Claudin-Low tumors. Thus, our data is not in disagreement with the previous conclusion from various groups for a basal origin of Claudin-Low tumors; our data suggest that loss of p53 in luminal cells facilitates luminal-to-basal cell fate change, and then Claudin-Low tumors may further develop from these intermediate basal-like cells. This may explain why even with a luminal origin, the majority of the resulting Claudin-Low tumors contained K14⁺ and/or K5⁺ basal-like tumor cells and almost all these tumors exhibited a basal mammary stem cell-like molecular signature. To make this point clear, we have revised the Abstract and the corresponding paragraph (Paragraph #4) in the Discussion accordingly, to indicate this potential tumor progression pathway.

REVIEWERS' COMMENTS:

Reviewer #4 (Remarks to the Author):

The manuscript has been improved with the addition of more tumor data. It would have been very instructive to take another luminal promoter such as WAP (using an adenoviral system since this is more rapid) to establish whether these findings are promoter- or model-specific (Point 1). I agree that a basal promoter-driven strain is not essential but confirmation of claudin-low tumors in the p53-deletion model using a different luminal promoter is important given the large variation seen with lineage tracing data. Comparative studies seem relevant to cell-of-origin investigations.

Responses to Reviewers' comments (Manuscript # NCOMMS-16-04284B):

REVIEWERS' COMMENTS:

Reviewer #4 (Remarks to the Author):

The manuscript has been improved with the addition of more tumor data. It would have been very instructive to take another luminal promoter such as WAP (using an adenoviral system since this is more rapid) to establish whether these findings are promoter- or model-specific (Point 1). I agree that a basal promoter-driven strain is not essential but confirmation of claudin-low tumors in the p53-deletion model using a different luminal promoter is important given the large variation seen with lineage tracing data. Comparative studies seem relevant to cell-of-origin investigations.

Response: We thank the Reviewer for carefully evaluating our revised manuscript and our response. Based on our experience, it appears mammary luminal cells targeted by the *Ad-Wap-Cre* adenovirus (using the *WAP* promoter) can also be targeted by the *Ad-K8-Cre* adenovirus. Nevertheless, we agree with this Reviewer that further comparative studies using a different luminal promoter would be necessary to fully support the conclusion of a luminal origin of Claudin-Low breast cancer. We have included this information in the Abstract and the Discussion section (page 17, first paragraph) of our revised manuscript.